# Independent replications reveal anterior and posterior cingulate cortex activation underlying state anxiety-attenuated face encoding
Sarah K. Buehler [1] ✉, Millie Lowther[1], Paulina B. Lukow[1], Peter A. Kirk[2], Alexandra C. Pike [3], Yumeya Yamamori[1], Alice V. Chavanne [4], Siobhan Gormley[5], Talya Goble[1], Ella W. Tuominen[1], Jessica Aylward[1], Tayla McCloud[1], Julia Rodriguez-Sanchez[1] & Oliver J. Robinson [1] ✉

Anxiety involves the anticipation of aversive outcomes and can impair neurocognitive processes, such as the ability to recall faces encoded during the anxious state. It is important to precisely delineate and determine the replicability of these effects using causal state anxiety inductions in the general population. This study therefore aimed to replicate prior research on the distinct impacts of threat-of-shock-induced anxiety on the encoding and recognition stage of emotional face processing, in a large asymptomatic sample ($n = 92$). We successfully replicated previous results demonstrating impaired recognition of faces encoded under threat-of-shock. This was supported by a mega-analysis across three independent studies using the same paradigm ($n = 211$). Underlying this, a whole-brain fMRI analysis revealed enhanced activation in the posterior cingulate cortex (PCC), alongside previously seen activity in the anterior cingulate cortex (ACC) when combined in a mega-analysis with the fMRI findings we aimed to replicate. We further found replications of hippocampus activation when the retrieval and encoding states were congruent. Our results support the notion that state anxiety disrupts face recognition, potentially due to attentional demands of anxious arousal competing with affective stimuli processing during encoding and suggest that regions of the cingulate cortex play pivotal roles in this.

The anxious state, characterized by an aversive anticipation of potential but unpredictable threats, involves increased vigilance and neurophysiological arousal (see Research Domain Criteria in *The National Institute of Mental Health Strategic Plan*[1]), as well as alterations in neurocognitive processing[2]. Importantly, this state can be adaptive by triggering appropriate defensive behaviours in uncertain environments[3,4], but may become pathological when exaggerated and chronic[5]. A better understanding of how adaptive anxiety alters neurocognitive processes can therefore inform our ability to detect and intervene early when this goes awry in pathological anxiety disorders. Yet, the precise functions and effects of state anxiety on cognition remain poorly understood. For instance, anxiety has been associated with abnormalities in the recognition of previously-seen emotional face stimuli[6–11]. This function is highly relevant since our environments are predominantly social and the ability to recognize familiar faces is pertinent

to navigating them successfully. It is also known that emotional faces are particularly salient stimuli, which benefit from enhanced processing over non-salient stimuli by preferentially recruiting attentional resources[12,13], and are prone to anxiety-related alterations[14–17]. However, empirical findings on the precise causal effects of state anxiety on face recognition are sparse and inconsistent in the experimental manipulations, task paradigms and stages of face memory processing investigated across different samples[6,11,18–20].

Two previous studies investigated behaviourally[7,8] as well as using functional magnetic resonance imaging (fMRI)[8] how a within-subjects threat-of-shock anxiety induction distinctly affects the encoding and retrieval stage of emotional face processing. Specifically, emotional face stimuli were presented in blocks of encoding followed by retrieval, which alternated between threat-of-shock and safe control conditions. The threat-of-shock manipulation reliably induces state anxiety through the

[1]University College London, London, UK. [2]National Institute of Mental Health, Washington, DC, USA. [3]University of York, York, UK. [4]University of Paris-Saclay, Paris, France. [5]University of Cambridge, Cambridge, UK. ✉e-mail: sarahkatharinabuehler@gmail.com; o.robinson@ucl.ac.uk

anticipation of unpredictable non-painful electrical shocks[21]. This promotes a range of physiological, neural and cognitive changes similar to those found in pathological anxiety[2,22] and has been proposed as an intermediate bridge between basic and clinical research[23]. Using this paradigm, both Bolton & Robinson[7] and Garibbo et al.[8], have found that the ability to recognize (i.e., retrieve) previously presented faces was reduced when they were perceived (i.e., encoded) under anxiety. At the neural level, Garibbo et al.[8] further demonstrated that this anxiety-attenuated face encoding was associated with increased activation in the anterior cingulate cortex (ACC), a cortical brain region that is, amongst other things, frequently implicated in cognitive functions that require attentional control[24,25].

The aim of the present study was to attempt in a large independent sample a direct replication of this threat-of-shock face recognition task by Bolton & Robinson[7] and Garibbo et al.[8], delineating the causal behavioural and neural effects of state anxiety on the encoding and retrieval of emotional faces. Given the increasing interest in anxiety disorder biomarkers and clinical relevance of anxiety-induction research, it is critical to determine the replicability of the neuroimaging and behavioural effects of within-subject state anxiety manipulations in non-symptomatic individuals before investigating how these differ in clinical populations. Over the past decade, a replication crisis has called into question the validity of findings from both psychological[26] and functional magnetic resonance (fMRI) neuroimaging studies[27]. In particular, studies with small sample sizes are often underpowered to detect reliable effects. Historically, the median sample size of experimental fMRI studies is less than 30[28], which can result in a substantial inflation of false-positive rates[29]. Substantially larger sample sizes and meta or mega-analytic approaches are required to ascertain the stability of effects across different samples. This highlights the importance of conducting direct replication studies in larger samples, which are statistically more powered to detect effects, ideally using pre-registered analysis plans to ensure transparency for future replications. With this in mind, we used the same experimental task and threat-of-shock manipulation administered by Bolton & Robinson[7] and Garibbo et al.[8] with a publicly pre-registered analysis plan, to test the replicability of the behavioural and neural effects in a large independent sample ($n = 92$).

Our primary analyses pertained to replicating the behavioural reduction in recognition accuracy for faces previously encoded during a threat-of-shock anxious state, and associated increase in brain activation in the ACC during the anxious face encoding but not retrieval stage. This hypothesized effect aligns with theoretical accounts that suggest state anxiety may compete for top-down allocation of attentional resources, which could be signalled by increased ACC activation, and that face identity encoding may be particularly vulnerable to this disruption[13,30,31]. In this replication study, we sought to clarify if this behavioural effect emerges reliably in a large independent sample and with neural activation in the same ACC region. We therefore used the ACC activation from Garibbo et. al as a region of interest (ROI) alongside a whole-brain analysis, since cingulate regions form part of a wider neural network involved in attentional and anxiety-related processes[2]. In our secondary neural analysis, we further hypothesized to replicate an increase in activation in the bilateral hippocampus when the state, threat-of-shock or safety, during face retrieval is congruent with the state during prior encoding (ROI from Garibbo et al.[8]). This aligns with extensive evidence implicating the hippocampus in the reactivation of context-item associations formed during encoding[32], but remains to be replicated for this paradigm. Finally, we combined the original behavioural and neural data from Bolton & Robinson[7] and Garibbo et al.[8] with our sample to complement our primary and secondary analyses with a mega-analytic approach that indicates what effects are robust across the studies[33,34].

## Methods
### Participants
We collected data for a total sample of $n = 98$ participants, who completed the threat-of-shock face recognition task during a single fMRI scanning session between December 2017 and May 2022. This sample forms part of a larger pharmacological intervention study, but here we utilize only the data from asymptomatic individuals at baseline. Of those 98 individuals in our preregistration, five did not complete the task of interest so the final sample size for this study was $n = 93$ participants. The mean age was 24.22 years (SD = 7.35), 71% of the sample self-reported as female and 29% as male, with 49.5% identifying their ethnicity as Asian, 2.2% as Black, 41.9% as White and 6.5% as Mixed. For further analysis specific exclusions deviating from our initial preregistration see the methods subsections below. Participants were recruited using subject databases at University College London, Kings College London, the GLAD study (part of NIHR BioResource), MQ Participate and social media advertisements. During an initial screening, the Mini International Neuropsychiatric Interview (MINI; Sheehan et al.[35]) was administered to ensure no personal or family history of psychiatric disorders. Further exclusion criteria included: (i) general functional magnetic resonance imaging (fMRI) exclusions, (ii) general ill health, (iii) recent use of illicit drugs. All participants provided written informed consent and were reimbursed £7.50/hour for activities completed outside the scanner and £10/hour for scanning. Ethical approval was obtained from the UCL Research Ethics Committee (6198/002).

For additional meta-analytic analyses, we also collated data from two studies. These included $n = 86$ participants (50 females, 36 males, mean age = 24.7) who had completed the threat-of-shock face recognition task during a behavioural testing session (see ref. 7) and $n = 32$ participants (18 females, 14 males, mean age = 27.03) who had completed it during a fMRI scanning session (see ref. 8).

### Threat-of-shock face recognition task
This experimental task was identical to that in Garibbo et al.[8] (which was in turn replicated from Bolton and Robinson[7] and completed in an fMRI scanner. The task consisted of 4 blocks, each involving an encoding phase followed by a retrieval phase, alternating between a threat-of-shock state and safe control state (see Fig. 1a, b). This constituted a 2 (safe vs threat-of-shock) by 2 (encoding vs retrieval) design and resulted in the following combinations of encoding and retrieval respectively: safe followed by threat-of-shock, safe followed by safe, threat-of-shock followed by safe, threat-of-shock followed by threat-of-shock. A warning slide was presented (for 1.35 s) to inform participants which state they were entering, and a coloured frame reminded them throughout each condition (red for threat-of-shock, blue for safe). Each block contained a different set of 36 face stimuli (i.e., 144 in total) from the Chicago Face Database[36], with equal ratios of male to female as well as happy, fearful and neutral faces. Block order was counterbalanced across participants to account for shock desensitization over time and stimuli order was randomized within blocks. First, during encoding, 18 face stimuli were sequentially presented for 0.5 s each, separated by a fixation cross with an inter-stimulus interval between 0.75 and 2 s (ISI). After a longer fixation interval, between 7.5 and 12.5 s, participants entered the retrieval phase. During retrieval, 36 face stimuli were presented sequentially for 0.5 s, half of which were previously presented during encoding and the other half unseen. Following another ISI, participants were presented with a response slide asking whether they had seen the face before. This was on screen either until they responded by pressing a button corresponding to Yes or No (counterbalanced), or until more than 2 s passed, in which case an incorrect response was automatically recorded. Before and after the task a 30 s fixation was presented to provide an additional baseline for fMRI contrasts.

Prior to the task we followed a well-established shock work-up procedure outside the scanner[37,38], whereby different shock intensities (ranging between 1-10 mA) were tested to establish a level that was unpleasant but not painful for each participant. Electric shocks were administered using a Digitimer DS7 with two disk electrodes on participants' left ankle. The threat-of-shock paradigm induces anxiety through anticipation of unpredictable shocks, although unbeknownst to participants shocks are only delivered randomly 3 times during the entire task (restricted to threat-of-shock conditions).

### fMRI data acquisition

Scanning was conducted at the Birkbeck-UCL Centre for Neuroimaging (BUCNI) on a 1.5 T Siemens Avanto scanner, using a 32-channel head coil. A field map (T2*-weighted images: repetition time (TR) = 1170 ms, echo time (TE1) = 10 ms, TE2 = 14.76 ms, field of view (FOV) = 64 × 64, voxel size = 3 x 3 x 2 mm, slice thickness = 2 mm, flip angle = 90°, 64 volumes) was obtained whilst participants completed the training block, EPI scans (T2*-weighted images: repetition time (TR) = 3500 ms, echo time (TE) = 50 ms, field of view (FOV) = 64 × 64, voxel size = 3 mm, slices = 40, slice thickness = 2 mm, flip angle = 90°, approximately 228 volumes) were collected during the testing block, and an MPRAGE (T1-weighted images: TR = 2730 ms, TE = 3.57 ms, FOV = 224 × 256, voxel size = 1 mm, slice thickness = 1 mm, flip angle = 7°, 176 volumes) at the end of the session. Activities outside the scanner were completed at the UCL Institute of Cognitive Neuroscience.

### Data analysis

In our commitment to transparent and reproducible research, the analysis plan for this study was previously pre-registered on the Open Science Framework (see https://osf.io/6952a/), with a clear delineation of planned hypothesis-driven analyses as well as post-hoc deviations and exploratory analyses. Given the primary aim of this replication study, we focused here on the hypotheses and analyses required for replication purposes (detailed below). Other pre-registered analyses, such as face stimuli valence comparisons and functional brain connectivity analyses, will be reported in the supplementary material. We conducted all analyses using only open-source software (R and AFNI).

### Behavioural data analysis

Behavioural data analysis was conducted in R using two-tailed tests and considered significant at a threshold of $\alpha < 0.05$. To ensure replicability of the analyses by Bolton & Robinson[7] and Garibbo et al.[8] and given the negatively skewed distribution of our data, we deviated from the original preregistration and transformed (squared) the outcome measure (proportion of correct responses at retrieval) for all behavioural analyses. The untransformed data is displayed in the figures for visualization purposes (see supplementary fig. 1a, b *for transformed data plots*). To ensure that this data met the assumptions of the statistical test (ANOVA) used in our behavioural analysis, we ran Shapiro-Wilk normality tests using the *shapiro_test()* function in R to confirm there is no significant deviation from the normlal distribution in our model residuals for the effect of encoding state ($p = 0.58$) and retrieval state ($p = 0.72$). The *anova_test()* function in R from the *rstatix* package, which we used to compute the ANOVA models, automatically checks the assumption of sphericity internally using the Mauchly's test. From the original sample of 93, the behavioural data from one participant had to be excluded because no responses were recorded, so the sample size for all behavioural data analyses was $n = 92$. To ensure our results were not driven by participants who may have failed to understand or attend to the task, we reran our analyses excluding participants whose behavioural performance on the task was at or below chance level (mean recognition accuracy of $< 0.5$) across all four conditions. However, this only applied to 4 participants and did not change the significance of any effects.

### Primary behavioural hypothesis: reduced recognition of faces encoded under threat-of-shock compared to safety.

To address our primary behavioural hypothesis that threat-of-shock at encoding but not retrieval impairs face recognition accuracy, we statistically tested for a main effect of state (threat-of-shock or safety) at both the encoding and retrieval stage, as well as their interaction, using a within-subjects factorial analysis of variance (ANOVA). Specifically, the ANOVA framework allowed us to contrast face recognition accuracy for all state combinations of threat-of-shock at encoding (i.e., followed by either threat-of-shock or safety at retrieval) with safety at encoding (i.e., followed by either threat-of-shock or safety at retrieval), and for all combination of state at retrieval in the same manner. We deviated slightly from our preregistration by including the main effect of retrieval state and

the interaction between encoding and retrieval state in our model, to investigate if our hypothesized threat-of-shock effects are specific to the encoding stage.

### Exploratory behavioural hypothesis: improved recognition of faces retrieved in the same state (threat-of-shock or safety) as they were encoded in.

An additional within-subject ANOVA was run to investigate the effect of state congruency compared to incongruency on face recognition accuracy at retrieval. Specifically, the ANOVA framework allowed us to contrast face recognition accuracy at retrieval for all congruent state combinations (threat-of-shock encoding followed by threat-of-shock retrieval, safety encoding followed by safety retrieval) with all incongruent state combinations (threat-of-shock encoding followed by safety retrieval, safety encoding followed by threat-of-shock retrieval).

### Exploratory mega-analyses of behaviour across studies.

We also performed a mega-analysis, considered the 'gold-standard' of meta-analytic approaches and also referred to as individual-participant data (IPD) meta-analysis[33,34]. For this we combined the raw behavioural data from all participants ($n = 210$), across the current study (labelled as 'Buehler et al. 2024': $n = 92$) and the two previous studies utilizing the same threat-of-shock potentiated face recognition task (Bolton & Robinson[7]: $n = 86$; Garibbo et al.[8]: $n = 32$). Based on the aforementioned behavioural hypotheses, this was done to assess the robustness of (a) the effect of state (threat-of-shock or safety) distinctly at encoding and retrieval, as well as (b) the effect of state congruency at retrieval on face recognition accuracy across all three studies. We therefore used the same ANOVA models as in the primary and secondary analyses but accounting for study as a between-subjects factor. Of interest were the main effects of encoding state and retrieval state on recognition accuracy.

### fMRI data pre-processing

Before data analysis, the first four volumes were discarded to allow the magnetic field to stabilize. All fMRI data was pre-processed using the open source fMRIPrep pipeline version 20.2.7 (for full details see https://fmriprep.org/en/20.2.7/workflows.html). Anatomical pre-processing included skull stripping ('ANTs'), brain tissue segmentation (FSL's 'fast'), spatial normalization to standard MNI152NLin2009cAsym space ('ANTs'), surface reconstruction ('FreeSurfer'). This was followed by functional pre-processing, which included head-motion correction (FSL's 'mcflirt'), slice-time correction (AFNI's '3dTShift'), susceptibility distortion correction and co-registration of the functional EPI reference to the anatomical T1w image (FreeSurfer's bbregister). In addition to fMRIPrep, we spatially smoothed data to 6 mm FWHM, constrained within a MNI template grey matter mask (AFNI's '3dBlurToFWHM') and scaling of the timeseries in each voxel to a mean of 100 (AFNI's '3dTstat', '3dcalc').

### fMRI data analysis

The fMRI data analysis was conducted in R and AFNI (specific functions are denoted in parentheses). From the original sample of 93 (including the participant excluded from behavioural analysis), one subject had to be excluded due to scanner artifacts so the final sample size for the fMRI data analysis was $N = 92$.

### Within-subject modelling

For the within-subject modelling we constructed general linear models (GLMs) for each participant. With the regressors of interest we modelled the face onset times and duration (0.5 seconds) as events, with the fixation cross during the inter-stimulus intervals as well as start and end of task treated as an implicit baseline. This amounted to 8 regressors accounting for the face stimuli onsets during all state combinations, including 4 encoding phase regressors with 2 for stimuli during threat-of-shock encoding (i.e., once followed by threat-of-shock and once by safety at retrieval) and 2 for safety encoding (i.e., once followed by threat-of-shock and once by safety at retrieval), as well as 4 retrieval phase regressors for the same combinations of

state but with stimuli onsets during retrieval. These were convolved with the hemodynamic response function (approximated by a gamma function) using '3dDeconvolve' in AFNI. As nuisance regressors in all within-subject GLMs we further included movement-correction parameters (pitch, roll, yaw, z, y, z and derivatives of each motion type) as well as a parameter controlling for the time of shock delivery and presentation time of the safe/threat-of-shock warnings. To further control for motion artifacts, we censored volumes with framewise displacement exceeding 1.3 mm and excluded individuals with more than 20% of volumes requiring censoring (none in this sample).

To make inferences about the neural activation underlying the threat-of-shock potentiated face recognition task we assessed group-level effects in predefined regions of interest (ROIs) and at the whole-brain level.

### Group-level modelling

**Region of interest (ROI) Replication Analysis.** We investigated the replicability of the neural activation associated with our contrasts of interest in specific predefined regions of interest (ROIs). Specifically, we extracted the average beta coefficient values for the contrasts of interest across all voxels in the ROIs for every participant and subjected these to hypothesis testing in R. Due to redundancy, we did not include the results from another initially preregistered ROI analysis approach using small-volume correction.

**Primary neural hypothesis: increased ACC activation while encoding faces under threat-of-shock compared to safety.** To address our primary neural hypothesis that threat-of-shock at face encoding but not retrieval is associated with increased ACC activation, we used the resampled group-level dorsal ACC activation map (using AFNI's 3dmaskave) from Garibbo et al.[8] as an ROI for the following contrast of interest: faces during threat-of-shock blocks > safety blocks, at encoding and retrieval separately. To ensure consistency with the behavioural analysis, we then statistically tested for a main effect of state (threat-of-shock or safety) at both the encoding and retrieval stages, using a within-subjects factorial analysis of variance (ANOVA) of the extracted beta coefficients.

**Secondary neural hypothesis: increased hippocampus activation while retrieving faces under the same state (threat-of-shock or safety) as they were encoded in.** To address our secondary neural hypothesis that hippocampus activation is increased while retrieving faces under the same state (threat-of-shock or safety) as they were encoded in, we used the resampled bilateral hippocampus mask (using AFNI's 3dmaskave) previously extracted by Garibbo et al.[8] from the Wake Forest University PickAtlas toolbox[39] as an ROI for the following contrast of interest: faces during state congruent retrieval blocks (threat-of-shock/safety at encoding followed by threat-of shock/safety at retrieval) > incongruent retrieval blocks (threat-of-shock/safety at encoding followed by safety/threat-of shock at retrieval). To ensure consistency with the behavioural analysis, we then statistically tested for a main effect of state congruency using a within-subjects factorial analysis of variance (ANOVA) of the extracted beta coefficients.

**Exploratory mega-analyses of ROIs across studies.** As for the behavioural data, we followed up on the above primary and secondary neural analyses with a ROI mega-analysis. We did this by combining the beta weight coefficients from the within-subject modelling of our current sample with those available from Garibbo et al.[8] for a group-level analysis (total *n = 124)* for the contrasts of interest and ROIs specified in the aforementioned neural hypotheses. We then statistically tested for the same main effects using ANOVA models, as in our behavioural mega-analysis, by accounting for study as a between-subjects factor.

### Whole-brain analysis

Following ROI analyses, we conducted group-level whole-brain tests using AFNI's '3dMVM', which performs traditional ANOVA-style computations.

The beta weights for the regressors of interest from the within-subject modelling are provided to '3dMVM' and specified as within-subject regressors (using 'wsVars' option) and a one-way t-test applied to the contrasts of interest (using 'gltCode' option). We used a template MNI grey matter mask to constrain analyses. We accounted for family-wise errors using simulation-based cluster-correction, using AFNI's '3dClustSim' to estimate, based on simulations of false positive noise clusters (derived by '3dFWHMx' from spatial autocorrelation estimates of 3dMVM's group-level model residuals), the minimum required cluster size for a specified voxelwise-threshold of $p < 0.001$ and significance threshold of $p < 0.05$. We used bi-sided thresholding, whereby positive and negative values above the threshold are clustered separately, and AFNI's NN level 2, whereby clusters are defined when faces or edges touch. For these analyses we cannot report exact values for the t-statistic, $p$-value and confidence interval but only the minimum cluster size required.

### Exploratory analysis of activation when encoding and retrieving faces under threat-of-shock compared to safety

For the whole-brain analysis we specified the following contrast of interest: faces during threat-of-shock blocks > safety blocks, for encoding and retrieval separately. The beta values for the contrast of interest from significant clusters that emerged in the whole-brain analysis were extracted and plotted in the results sections. This was done for visualization purposes only and no statistical analyses were run on these.

### Exploratory mega-analyses of whole-brain activation across studies

We followed up on the encoding effect with a whole-brain mega-analysis by combining the beta weights from the within-subject modelling of our sample with those available from Garibbo et al.[8] for a group-level analysis (total $n = 124$) of the same contrast of interest: faces during threat-of-shock blocks > safety blocks, for encoding and retrieval separately. This was done to determine if both the anterior cingulate (ACC) cluster identified by Garibbo et al.[8] and the posterior cingulate cortex (PCC) cluster identified in our primary whole-brain analysis would emerge in the combined sample (voxel-wise threshold = $p < 0.001$, cluster-level significance threshold of $p < 0.05$). Note that to avoid double dipping we did not use the mega-analysis for the identification and discussion of new clusters. We specified the group-level ANOVA model (*using AFNI's 3dMVM*) in the same manner as in our behavioural mega-analysis, by accounting for study as a between-subjects factor.

### Reporting summary

Further information on research design is available in the Nature Portfolio Reporting Summary linked to this article.

## Results

In this study we utilized the same threat-of-shock face recognition task previously investigated in a behavioural study by Bolton & Robinson[7] and behavioural fMRI study by Garibbo et al.[8] to determine the replicability of the impact of induced anxiety on the encoding and recognition of emotional faces. The task consisted of 4 blocks, each involving an encoding condition followed by a retrieval condition, alternating between a threat-of-shock state and safe control state (see Fig. 1a, b). To ensure consistency with the analyses by Bolton & Robinson[7] and Garibbo et al.[8] and given the negatively skewed distribution of our data, the outcome measure (proportion of correct responses at retrieval) was transformed (squared) for all behavioural analyses (for transformed data plots see supplementary fig. 1a, b).

### Behavioural effects of threat-of-shock on face recognition

To determine if threat-of-shock at encoding but not retrieval impairs face recognition in our sample, we tested for a main effect of state (threat-of-shock or safety) at the encoding and retrieval stage, as well as their interaction on face recognition accuracy using a within-subjects factorial analysis of variance (ANOVA). In our sample ($n = 92$) we replicated a significant

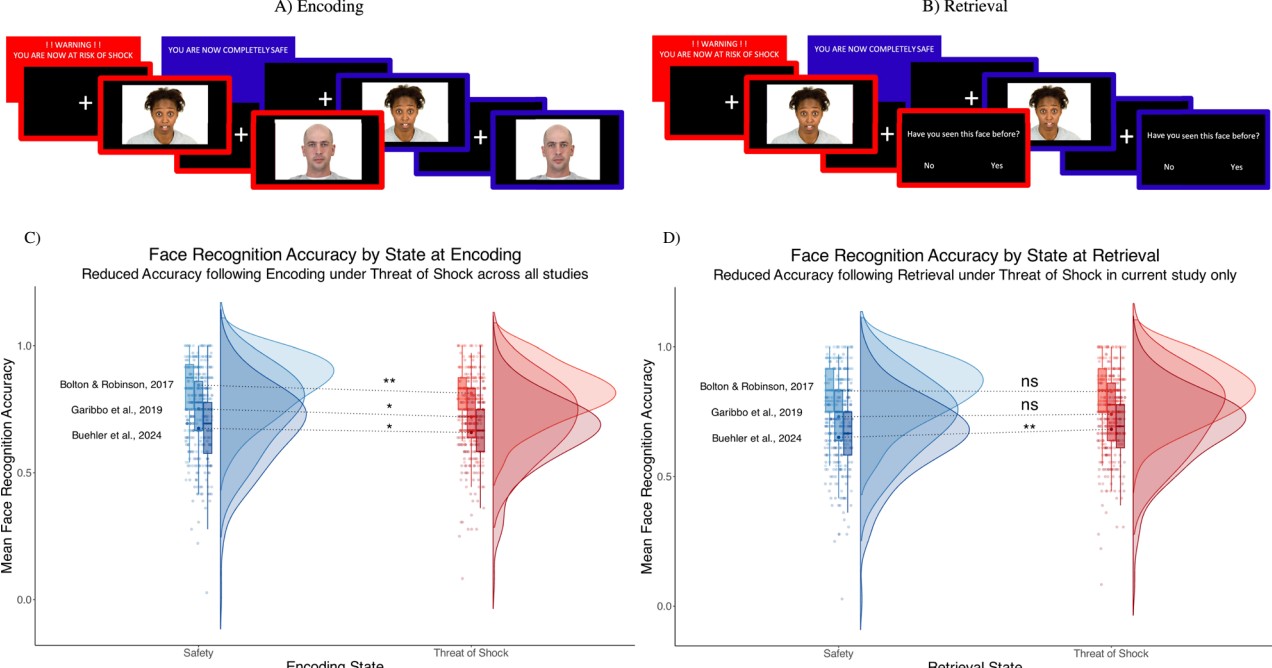

**Fig. 1 | Across independent studies threat-of-shock during encoding impairs face recognition accuracy.** The threat-of-shock face recognition task consists of four blocks with an encoding condition (**A**) followed by a retrieval condition (**B**), alternating between a threat-of-shock state (red) and safe control state (blue). A warning slide informs participants which state they are entering, and a coloured frame reminds them throughout (red for threat-of-shock, blue for safe). Emotional face stimuli are presented during encoding (18 unseen) and retrieval (18 seen, 18 unseen), following which participants are asked if they have seen the face before and respond (yes/no) with a button-press. In our behavioural analysis of this task we combined the current study (labelled 'Buehler et al. 2024': n = 92) with data from two previous studies[7,8]. This indicated that (**C**) recognition accuracy is significantly impaired when faces were previously presented under threat-of shock (red) compared to safety (blue) during encoding, in our study and a mega-analysis across all three studies. But during (**D**) retrieval there is a significant improvement in retrieving faces under threat-of-shock in our study only, which was not, however, statistically significant in a mega-analysis across all three studies.

main effect of encoding state ($F = 4.247$, $df = 91$, $p = 0.042$, $\eta_p^2 = 0.045$, see Fig. 1c), with reduced mean face recognition accuracy at retrieval when faces were encoded in a state of threat-of-shock (Mean=0.659, SD=0.142) compared to safety (Mean = 0.675, SD = 0.158). We also found a significant main effect of retrieval state ($F = 10.507$, $df = 91$, $p=0.002$, $\eta_p^2 = 0.104$, see Fig. 1d), with higher mean accuracy achieved when retrieving faces under threat-of-shock (Mean = 0.683, SD = 0.147) than safety (Mean = 0.652, SD = 0.152). There was no significant interaction between encoding and retrieval state ($F = 0.482$, $df = 91$, $p = 0.489$, $\eta_p^2 = 0.005$).

We then ran an individual participant data (IPD) mega-analysis across the behavioural data from the two previous studies (Bolton & Robinson[7]: n = 86; Garibbo et al.[8]: n = 32) and current study (labelled as 'Buehler et al. 2024': n = 92), to determine what effects are robust in a combined sample (n = 211). This revealed a significant main effect of state on face recognition for encoding ($F = 16.777$, $df = 207$, $p < 0.001$, $\eta_p^2 = 0.075$) but not retrieval ($F = 3.060$, $df = 207$, $p = 0.082$, $\eta_p^2 = 0.015$), while accounting for study as a between-subjects factor (encoding state*study interaction: $F = 0.968$, $df = 207$, $p = 0.382$, $\eta_p^2 = 0.009$, retrieval state*study interaction: $F = 3.195$, $df = 207$, $p = 0.043$, $\eta_p^2 = 0.03$).

**Neural activation underlying face encoding under threat-of-shock**

To address our primary neural hypothesis, we investigated the replicability of enhanced activation when faces were encoded or retrieved under threat-of-shock compared to safety using the group-level dorsal ACC activation map from Garibbo et al.[8] as a ROI as well as using an exploratory whole-brain analysis of our sample (n = 92). In the anterior cingulate cortex (ACC) ROI in our sample we found no significant difference in neural activation when comparing threat-of-shock to safety during encoding ($F = 1.208$, $df = 91$, $p = 0.275$, $\eta_p^2 = 0.013$) or retrieval ($F = 1.416$, $df = 91$, $p = 0.237$, $\eta_p^2 = 0.015$). However, the whole brain analysis of our sample revealed

significant activation when comparing threat-of-shock to safety during encoding, but not retrieval, in a posterior cingulate cortex (PCC) cluster (size: 43 voxels, peak: x = +0.5, y = +38.5, z = +49.5, centre of mass: x = +0.4, y = +32.7 z = +44.3, see Fig. 2a for group-level cluster and Supplementary Fig. 3a for visualisation of coefficients).

We then followed up on this with an individual participant data (IPD) exploratory mega-analysis of the ACC ROI and whole-brain to determine if activation in these brain regions emerges in a more powered analysis across samples. For this we combined the beta weights from the within-subject models for the same contrast of interest from Garibbo et al.[8] with our sample (total N = 123), accounting for study as a between-subjects factor. In the ACC ROI, there was a significant main effect of encoding state, with increased activation when faces were encoded under threat-of-shock compared to safety ($F = 55.921$, $df = 121$, $p<0.001$, $\eta_p^2 = 0.316$) in the combined sample (encoding state*study interaction: $F = 48.603$, $df = 121$, $p<0.001$, $\eta_p^2 = 0.287$). We found no statistically significant increased BOLD response for retrieval state ($F = 2.003$, $df = 121$, $p = 0.16$, $\eta_p^2 = 0.016$) in the combined sample (retrieval state*study interaction: $F = 3.243$, $df = 121$, $p = 0.074$, $\eta_p^2 = 0.026$). The whole-brain analysis further revealed that both the PCC cluster (size: 274 voxels, peak: x = −1,5, y = +28.5., z = +43.5, centre of mass: x = +3.7, y = +27.1., z = +43.8) and ACC cluster (size: 1009 voxels, peak: x = +0.5, y = -45.5, z = +21.5, centre of mass: x = +1.7, y = -45, z = +17.8) emerged when contrasting threat-of-shock with safety at face encoding, but not retrieval, in the combined sample at a voxel-wise threshold of $p < 0.001$ and cluster-level significance threshold of $p < 0.05$ (see Fig. 2b for group-level cluster and Supplementary Fig. 3b, c for visualisation of coefficients).

**Neural effects of state congruency at retrieval in hippocampus**

Using an ROI analysis to address our secondary neural hypothesis, we also found evidence for a significant increase in average neural activation in

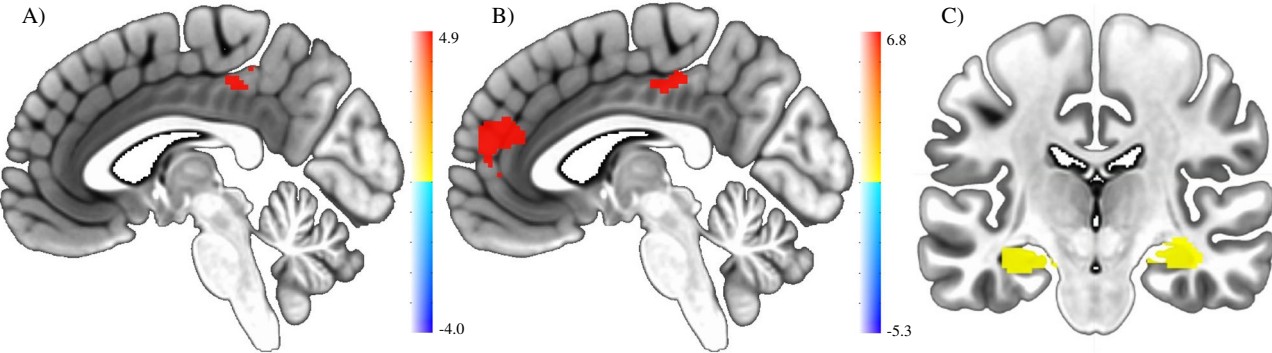

**Fig. 2 | Enhanced neural activation in anterior and posterior cingulate cortex underlies face encoding under threat-of-shock.** Neural Activation during the threat-of-shock face recognition task showing that (**A**) a posterior cingulate cortex cluster (PCC; see thresholded group-level cluster on MNI template brain with t-statistic bar) was significantly more active when encoding faces under threat-of-shock compared to safety in a whole-brain analysis of the current study. Further, when combining the current study and within-subject results from Garibbo et al.[8] in a whole brain mega-analysis, we found that (**B**) significantly enhanced neural activation while encoding faces under threat-of-shock compared to safety was evident in both the anterior cingulate cortex (ACC) and posterior cingulate cortex (PCC). In addition, congruency in the state (threat-of-shock or safety) between the encoding and retrieval phase was associated with (**C**) increased activation in a bilateral hippocampus ROI when the state during retrieval was congruent compared to incongruent with encoding in the current study ('Buehler et al. 2024': n = 92) and a mega-analysis combining the current study with Garibbo et al.[8]. See Supplementary Fig. 3a–c for visualisation of the model coefficients.

the bilateral hippocampus when the state (threat-of-shock or safety) during retrieval was congruent with encoding (*see* Fig. 2c *for group-level ROI cluster and* Supplementary Fig. 4b,c *for visualisation of coefficients*). This was reported in the original study (see Garibbo et al.[8]) and replicated in our sample ($F = 7.144$, $df = 91$, $p = 0.009$, $\eta_p^2 = 0.073$), as well as in a combined ROI mega-analysis (congruency main effect: $F = 13.534$, $df = 121$, $p < 0.001$, $\eta_p^2 = 0.101$, congruency*study interaction: $F = 1.586$, $df = 121$, $p = 0.21$, $\eta_p^2 = 0.013$).

## Discussion

In this study, we attempted to replicate the effect of experimentally induced state anxiety on the encoding and recognition of emotional faces[7,8]. We replicated a behavioural impairment in recognizing faces that were previously encoded under threat-of-shock, as was found in the prior studies. Our analysis of the underlying neural activation revealed that the enhanced ACC activity previously found by Garibbo et al.[8] during face encoding under threat-of-shock did not replicate in an ROI analysis in our sample. But a whole-brain analysis revealed a posterior cingulate cortex PCC cluster for this same contrast and, critically, both regions (ACC and PCC) emerged in a whole brain mega-analysis across studies. Finally, we replicated that retrieving faces in a state (threat-of-shock or safety) congruent with encoding was also associated with enhanced activation in the hippocampus.

The impaired recognition of faces encoded during anxiety found across all studies provides support for the notion that the anxious state may disrupt face recognition by interfering distinctly with the initial processing (i.e. encoding) stage[19,40]. This is consistent with other literature reporting anxiety-related difficulties in matching faces (Attwood et al.,[6] or recognizing facial features, such as emotional expressions[41] or other identifying features[42], even in the absence of memory. Face encoding is thought to be a largely perceptual process, which may be highly susceptible to disruptions by threat-of-shock-elicited physiological arousal[43,44]. Consistent with proposals of selective attention, more attentional resources may be allocated to task-unrelated threat-signals during perceptual processing, thus reducing attention allocated to face encoding[13,45–47]. In line with other work on dual process competition it is specifically in the case of task-irrelevant anxiety, which we were interested in eliciting by our threat-of-shock paradigm, that arousal signals compete with cognitive processes, so this may be unaffected or reversed when the source of threat is task-relevant[22,48,49]. Although there was evidence for an improvement in recognition accuracy during threat-of-shock retrieval in the present sample, this was not statistically significant in the previous two studies[7,8] or combined mega-analysis. It could be that memory processes at retrieval do not facilitate or even reduce the negative

impact of threat-signalling distractors. These results align with two-component models of the effect of anxiety on cognition, which suggest an inflection point where anxious arousal can impair processing when external task demands are low, but may not affect or even improve performance when more executive resources are required, such as during working memory retrieval[50]. At a mechanistic level, the observed impairment may have occurred as a result of more attentional resources allocated away from the perceptual processing of external face stimuli and towards the highly salient threat-signal. While we ensured our results were not driven by participants who did not attend to the task at all (see behavioural analysis in methods section), it could be that these attentional lapses occurred on a trial basis throughout the encoding stage. It is also possible that the disruption to face encoding during threat-of-shock relates more specifically to an impairment in the formation of accurate representations of face identity or committing of perceptual content to short-term memory for later retrieval.

The neural activation associated with this anxiety-related behavioural impairment may help clarify these potential underlying mechanisms. As demonstrated in the mega-analysis, both the ACC and PCC appear to be implicated in face encoding under threat-of-shock. While increased activation in the ACC has been reported across a range of task-based and anxiety-potentiated paradigms[51–54], the PCC is more commonly associated with the default mode network and introspective resting states[55–57], and its involvement in this task may have only emerged due to the large sample size. Nevertheless, both regions are part of the cingulate cortex and map onto a range of cognitive functions, such as the processing of faces and other affective stimuli[31,52,58,59]. Previous evidence also suggests that both anterior and posterior regions of the cingulate respond to unpredictable threats[60,61] and may be part of a wider anxiety-related network[2]. Alongside reciprocal connections to prefrontal and subcortical structures, such as the amygdala and hippocampal system, there is further support for the role of ACC and PCC in modulating attentional control and memory storage for salient stimuli[62–64]. This is consistent with the idea that preferential activation in these regions might signal the conflicting attentional demands of the internal anxious state at the expense of external face stimuli encoding. Specifically, activity in the posterior and anterior cingulate regions (as well as the precuneus, which overlaps with our observed PCC cluster) has been shown to signal the tuning of selective attention to internal, self-relevant cues and increase with arousal state[63,65–69]. For instance, increased activation in these regions has been observed when participants selectively attended to their internal emotional response to affective stimuli[70,71] and during the up- and down-regulation of negative emotions[72]. In line with the influential attentional control theory, which posits that anxious arousal reduces the

attentional processing of external stimuli[30], there may be an increase in internally directed attention to one's physiological response under threat-of-shock. This, in turn, could be signalled by increased anterior and posterior cingulate cortex activation in the brain and impair the ability to accurately encode face stimuli in our task.

During retrieval on the other hand, we found an increase in hippo-campal engagement in our task and across studies when the state (threat-of-shock or safety) was congruent to that experienced during encoding, despite an absence of statistically significant evidence for a behavioural effect across studies. This is consistent with the idea that this state-congruent hippo-campus activation may signal the recollection of context rather than the presented stimuli[73,74]. Much evidence points to the hippocampus' involvement in the retrieval of context-cue associations, such as shock-stimuli, where it reflects a reactivation of the neural representation of the context experienced during encoding[75,76]. It is a crucial area for relaying contextual signals to the amygdala and a broader prefrontal network, making it particularly relevant for threat-related processing in both adaptive and maladaptive anxiety[77]. Its causal role has been established by studying the effects of hippocampal lesions, which have been found to impair the contextual reinstatement of conditioned threat responses[78]. Animal work, which has since been replicated in humans[79], has also experimentally demonstrated disrupted context-dependent retrieval of threat-related memories when the hippocampus was inactivated via GABA receptor agonism[80]. This suggests that the hippocampal system may play a necessary, automatic, and adaptive role in maintaining emotionally well-regulated context associations. As such, the state-congruent activity increase we observed in the hippocampus during retrieval may be considered a naturally adaptive neural response that supports the retrieval of contextual cues rather than the specific face stimuli.

These results highlight the importance of conducting fMRI replication studies. Using the same behavioural paradigm, we found consistent anxiety-induced behavioural impairments across three independent studies. Yet, the precise brain activation associated with the same behavioural effect differed in our independent sample. To determine what is consistent and replicable across studies, mega-analyses of behavioural and neural responses can be a powerful tool, especially for closely replicated paradigms. This approach allowed us to demonstrate that face encoding may be associated with neural activity in *both* the anterior and posterior cingulate cortex. It also revealed replicable activation in hippocampus signalling when the context (threat-of-shock or safety) during retrieval was congruent to encoding. Replicating identical task paradigms uniquely allows for the identification of different brain regions that may be involved in related cognitive processes[81]. But given the current scarcity of fMRI replication studies it remains unclear how common such variability is. This has important implications for our understanding of anxiety disorders and their treatment response. Specifically, it is crucial to determine the replicability of behavioural and neuroimaging effects stemming from state anxiety alterations in non-clinical samples, to help ensure future comparisons with patient populations reflect commonalities and differences that are clinically meaningful rather than driven by random variability from unreliable tasks whose effects fail to replicate[2].

### Limitations
There remain several conceptual and methodological limitations, as well as suggestions we provide for future research. Firstly, while our main result pertains to the effect of threat-of-shock on face encoding, the behavioural outcome has to be measured at retrieval, so it remains unclear whether the impairment arises at the stage of initial processing or transition to short-term memory. For this, future work is needed to develop and validate direct measures of encoding (e.g., face specific neural replay signatures). Secondly, it remains unknown if the behavioural and neural effects we found are specific to faces or would also extend to other emotionally salient and non-salient stimuli. It is plausible, for instance, that effects are driven by features of the stimuli beyond the social and emotional relevance (e.g. the shape, colour or other generic feature). Future work is also needed to clarify underlying mechanisms, such as probing the potential role of attentional

shifts to interoceptive signals directly (e.g., subjective report, cues to focus attention on task) or indirectly (e.g., online measurement of the heartbeat-evoked potential). This would enable us to determine if the mechanism of impairment is indeed due to attention being directed away from the encoded stimuli. Future variations of this task could also utilize other threat-unrelated control conditions, such as the chance to win money for accurately retrieved faces. We also note that in our fMRI mega-analysis the within-subject models from Garibbo et al.[8] had been pre-processed and analysed in SPM rather than AFNI and are therefore subject to any discrepancies between these software programmes. Finally, while well-powered to detect the neural effects we were interested in, our task's block design (e.g., short stimulus presentations and inter-trial jitter) was not optimized to tease apart the contribution of individual face stimuli features such as valence, due to inherent delays in the hemodynamic response function (*see supplementary materials on face valence*). This also limited our ability to conduct more fine-grained analyses, such as investigating if 'successful' retrieval of correctly remembered faces parametrically modulates neural responses. However, these paradigm variations would not be consistent with the aims of this current study and warrant future research.

### Conclusions
In sum, this study provides evidence for a replicable impairment in emotional face recognition following induced anxiety during the initial processing stage of encoding, across three independent samples. Mechanistically, this could be due to shifts in attentional allocation, for instance, away from external face stimuli processing towards regulating internal anxious arousal. This may be signalled by the associated increase in neural activation observed in both anterior and posterior regions of the cingulate cortex. Alongside this, enhanced hippocampal activation may signal contextual congruency with the encoding state during subsequent retrieval. Together, these results suggest that alterations in emotional face encoding may be a stable cognitive signature of induced state anxiety.

### Data availability
The fully anonymized and non-identifiable behavioural data that support the main findings of this study, including data from the current sample as well as previous studies analysed[7,8], are publicly available on OSF: https://osf.io/6952a/. The group-level statistical maps for fMRI data that support the main findings of this study are publicly available on Neurovault: https://neurovault.org/collections/16528/. Individual participant fMRI data are not openly available as they cannot be anonymised but can be made available from the corresponding author upon request and completion of a data sharing agreement.

### Code availability
The code used for the behavioural analysis is publicly available alongside the behavioural data on OSF: https://osf.io/6952a/. The individual-level participant fMRI can be made available upon request and completion of a data sharing agreement.

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

## Acknowledgements

The authors would like to thank Dr. Rick Adams, Dr. Harry Costello and Dr. Karel Keislich for performing clinical cover over the study duration, as well as Aoife Fitzgerald, Elizabeth Long, Mackenzie Murphy, Ariana Reichler, Meghna Julien-Sebastien and, Giulia Spaeth for their help with participant recruitment. This research was funded by a UK Research and Innovation Medical Research Council senior nonclinical fellowship to O.J.R. (MR/R020817/1) and supported by the Biotechnology and Biological Sciences Research Council [grant number: BB/T008709/1] Ph.D. Studentship to S.B., The funders had no role in study design, data collection and analysis, decision to publish or preparation of the manuscript. We thank the NIHR BioResource Centre Maudsley and the NIHR BioResource volunteers for their participation, and gratefully acknowledge NIHR BioResource centres, NHS Trusts and staff for their contribution. The views expressed are those of the authors(s) and not necessarily those of the NHS, the NIHR or the Department of Health and Social Care.

## Author contributions

S.K.B. wrote the paper, completed the data analysis and contributed to the data pre-processing. M.L. contributed to the recruitment, screening, scanning and data pre-processing. P.B.L. contributed to the data pre-processing. P.A.K. contributed to oversight of data pre-processing and analysis, A.C.P. contributed to the recruitment, screening and scanning. Y.Y. contributed to the recruitment, screening and scanning. A.V.C. contributed to the recruitment, screening and scanning. S.G. contributed to the setup, recruitment, screening and scanning. T.G. contributed to the recruitment, screening and scanning. E.W.T. contributed to the recruitment, screening and scanning. J.A. contributed to the setup, recruitment and screening. T.M. contributed to the recruitment, screening and scanning. J.R. contributed to the recruitment, screening and scanning. O.J.R. contributed to writing the paper, the setup and oversight of recruitment, screening, scanning, data pre-processing and data analysis.

## Competing interests

The authors declare the following competing interests: Oliver J. Robinson (O.J.R.) ran an investigator-initiated trial with medication donated by Lundbeck (escitalopram and placebo, no financial contribution). He also held an MRC-Proximity to discovery award with Roche and has completed consultancy work for Peak, IESO digital health, Roche and BlackThorn therapeutics. O.J.R. sat on the committee of the British Association of Psychopharmacology until 2022. All other authors declare no competing interests.
