## [Peer Review File · Communications Psychology]

10th Apr 24

Dear Ms Buehler,

Thank you for your patience during the peer-review process. We are sorry for the significant delay in returning to you with a decision, which resulted from issues with reviewer availability. Your manuscript titled "Investigating the replicability of neural mechanisms underlying state anxiety-attenuated encoding of emotional faces" has now been seen by three reviewers, whose comments are appended below. You will find two reports below, as report #2 was co-authored by two referees working together.

You will see that they find your work of some potential interest. However, they have raised quite substantial concerns that must be addressed. In light of these comments, we cannot accept the manuscript for publication, but would be interested in considering a revised version that fully addresses these serious concerns.

We hope you will find the Reviewers' comments useful as you decide how to proceed. Should additional work allow you to address these criticisms, we would be happy to look at a substantially revised manuscript. If you choose to take up this option, please highlight all changes in the manuscript text file, and provide a detailed point-by-point reply to the reviewers.

Editorially, we would like you to improve the clarity in the theoretical conceptualization, and to elaborate on the rationale for both the meta- and mega-analyses. Relatedly, we would like to emphasize a number of presentational/statistics issues that are also captured by the reviewer team (Refs #2): Following our guidelines for statistics reporting and interpretation, please do not interpret non-significant results in null-hypothesis significance testing in the absence of positive evidence for the null from equivalence tests or Bayesian statistics. Claims of specificity, for example, also require appropriate interaction contrasts and positive evidence for the null. Moreover, as highlighted by Reviewer(s) #2, claims of specificity are also contingent on appropriate design choices, such as necessary control conditions. Where either of these components is lacking, these claims must be removed, or caveated. For example, there is currently no positive statistical evidence for the absence of a difference in face recognition accuracy between threat-of-shock and safe control condition at retrieval in the meta-analysis; this interpretation relies on a non-significant finding derived from NHST, which means that the claim of specificity of the difference to encoding cannot be maintained.

Please also remove any strong claims of clinical relevance or biomarker framing, we discourage these kinds of inferences based on work with healthy populations.

I am attaching a checklist that details critical reporting requirements for the revised manuscript. Please attend to each item and ensure your manuscript is fully compliant. We are requesting that your manuscript aligns with these requirements as this facilitates the evaluation of your manuscript, reducing delays in re-review and potential future acceptance. If your revised manuscript is not aligned with these requests on major issues, such as those concerning statistics, it may be returned to you for further revisions without re-review. Additional information can be found in our style and formatting guide Communications Psychology formatting guide.

If the revision process takes significantly longer than five months, we will be happy to reconsider your paper at a later date, provided it still presents a significant contribution to the literature at that stage.

Please use the following link to submit your

- revised manuscript,
- point-by-point response to the referees' comments,
- cover letter (as a separate document),
- the Editorial Policy Checklist (see below),
- the Reporting Summary (see below), and
- the completed Editorial Request Table (attached):

[link redacted]

Thank you for the opportunity to review your work.

Best regards,

Xiaoqing Hu

Xiaoqing Hu, PhD

Editorial Board Member

Communications Psychology

orcid.org/0000-0001-8112-9700

REVIEWER EXPERTISE:

Reviewer #1 anxiety, fMRI, computation modeling

Reviewer #2 affective neuroscience, fMRI, computation modeling

REVIEWER REPORTS:

Reviewer #1 (Remarks to the Author):

The study investigated whether an induced anxious state in healthy individuals during encoding and retrieval would lead to impairments in emotional face recognition and the corresponding neural correlates using fMRI. Specifically, the study aimed to evaluate the replicability of previous findings by using a new sample, as well as conducting a mega- and meta-analysis combining three independent samples. The results showed relatively strong evidence supporting the idea that an

anxious state during encoding impaired subsequent face recognition. Partially consistent with earlier findings, these impairments were shown to be related to activity in the cingulate cortex. I find this study to be well-designed and to demonstrate an efficient combination of existing data and new data to address an important question in affective science and cognitive science. The findings from this study enhance our knowledge about the neural mechanisms underlying the influence of anxiety on cognitive functioning and have potential implications for anxiety research. The threat-of-shock task used to induce an anxious state is appropriate, and the data analyses are well-suited for addressing the target questions. Below, I have a few major and minor comments, which I believe can be addressed through revision.

Major:

1. My major concern is about the clarity in the theoretical conceptualization. Specifically, in the introduction and discussion, the authors seem to imply that attenuated face processing during an anxious state is related to ACC activity, and ACC is known to be related to attention control. Therefore, the attenuated face processing is related to reduced attention allocation during an anxious state. Further, the authors seem to imply that the task evoked anxious apprehension, which is the cognitive dimension of anxiety, but the attention demand is because of the anxious arousal, which is the physiological dimension of anxiety. If this is what the authors meant to communicate, then I would suggest making the reasoning more explicit. However, it remains unclear how anxious apprehension, the cognitive dimension of anxiety, leads to the attention allocation difficulty related to anxious arousal, the physiological dimension.

2. Relatedly, the theories driving the original hypotheses are unclear to me. I hope the authors can further elaborate and/or clarify the theoretical advancements gained from these findings. In addition, the authors may add implications on emotional control.

3. While the authors claimed that part of the motivation to conduct this study is the clinical relevance, specifically, it is important to determine the replicability of neuroimaging and behavioral effects in the general population before investigating the individual differences related to clinical conditions. I highly agree with the authors on this point of establishing within-subject effects in the general population first. But I encourage the authors to point out that even a robust within-subject effect does not warrant individual differences.

4. Regarding the task, the face stimuli included emotional faces. However, it is unclear what the rationale is for including emotionality in this task in the present paper.

5. fMRI methods. Please clarify if this task is a single scan session task or multiple sessions, with each corresponding to a block.

6. The discussion highlighted the consistency between the current PCC and the previous ACC findings. The authors could further elaborate on the potential differential roles. This may have different theoretical implications.

Minor:

1. In the introduction, the authors may consider citing APA and RDoC works when defining anxiety.

2. Also in the first paragraph, references seem to be needed when the authors wrote that inconsistent findings exist across samples and different paradigms.

3. In the methods section, it would be helpful to briefly describe the samples of the prior studies in the present article for ease of reading.

4. The Meta- and Mega-analysis methods section seems to be under-referenced.

Reviewer #2 (Remarks to the Author):

#####

Investigating the replicability of neural mechanisms underlying anxiety-attenuated encoding of emotional faces

Buehler...& Robinson

#####

The present study leverages a relatively large sample to replicate and extend prior work on the impacts of threat-of-shock-induced anxiety on the encoding and recognition stage of emotional face processing. The authors conclude that anxiety impairs face recognition, possibly due to attentional interference during encoding, and that the cingulate cortex is critical to this process.

Key Strengths (Significance, Innovation, Rigor, Approach):

large pooled sample

attempt to grapple with the replication crisis

pre-registered approach

clear articulation of the gap that the present work seeks to address

thoughtful acknowledgement of key limitations in discussion

Key Weaknesses (Significance, Innovation, Rigor, Approach):

as written, the rationale for the inclusion of both the meta- and mega-analyses is undermotivated in the introduction

relatedly, the authors do not have a clear, consistent approach for the analyses across studies/samples...authors might consider either doing replication analyses for behavior only or just doing the same thing for behavior and for brain metrics

need for a more balanced discussion (not overselling clinical relevance/use of biomarkers/limitations based on sample composition and study design)

In the section that follows, I provide a few suggestions for further strengthening the manuscript.

MAJOR / GENERAL

1. INTRODUCTION.

-As written, the rationale for the inclusion of both the meta- and mega-analyses is undermotivated in the introduction. While both mega- and meta- analytic approaches are an important step forward, it is unclear whether both are necessary in the present context. I encourage the authors to consider the complimentary strengths and weaknesses of including both. Perhaps they are sufficiently redundant so as to scrub the meta-analysis.

-“Given the clinical relevance of this research, such as the potential development of anxiety-disorder biomarkers, it is critical to first determine the replicability of neuroimaging and behavioural effects pertaining to induced anxiety-cognition interactions in the general population. For instance, if unrelated brain areas stochastically emerge for the same contrast across different healthy samples, they would have no value as pre-clinical biomarkers. Instead, substantially larger sample sizes or meta-analyses may be required to ascertain the stability of evoked activations.”

-Some of the aspirational clinical significance noted in the introduction seems far-fetched given that this is a super healthy (asymptomatic with no lifetime family history) sample using a threat-of-shock task. This isn't a population representative sample.

-Additionally, I encourage the authors to consider minimizing or eliminating the biomarker content, as it deters from the real strengths of the paper (e.g., large N, pre-registered approach). It is not clear what type of biomarker this work would yield (e.g., case-control biomarker, one relevant for treatment, or one for identifying risk?), and it is further unclear whether a biomarker derived from this healthy sample would prove useful.

2. METHOD.

- As noted above, some of the aspirational clinical significance and biomarker language noted in the introduction seems far-fetched given that this is a super healthy (asymptomatic with no lifetime family history) sample using a threat-of-shock task. This isn't a population representative sample. Careful re-consideration of this language will allow the genuine strengths of the paper to stand out.

3. ANALYTIC STRATEGY.

4. RESULTS.

-The paper would benefit from clearly distinguishing hypothesis testing from exploratory testing. For example, in the introduction, the authors hypothesize that “anxious encoding would be associated at the neural level with increased activation in the ACC.” However, in the data analysis section of the method, they write, “[w]e used the resampled group-level dorsal ACC activation map (using AFNI's 3dmaskave) from Garibbo et al., (2019) as an ROI for the following contrast of interest: threat-of-shock > safety, during encoding and retrieval separately.” It is unclear to the reader why retrieval is being examined when the stated hypothesis pertains to encoding.

-Moreover, a consistent use of nomenclature would aid the reader in following along. THTH versus TofS becomes somewhat difficult to keep straight. Why not just consistently use Threat vs Safety?

-Another way to compare threat and safe would be to compare TS and TT to SS and ST...the primary behavioral hypothesis is framed in ANOVA language, but as written, it is not clear that ANOVA is the most appropriate overarching analytic framework. Further clarity around how contrasts were set up would greatly aid the reader.

-Likewise, for congruence is it TT and SS vs. ST and TS (i.e., congruent vs incongruent)? Stating the contrasts more explicitly would enhance the report.

-Put differently, the authors might consider adopting theory-driven focused contrasts (e.g., encode under threat/retrieve under threat and encode under safety/retrieve under safety compared to encode under safety/retrieve under threat and encode under threat/retrieve under safety). This may be clearer than using ANOVA main effect language since the results are not really main effects in the traditional way.

5. DISCUSSION / IMPLICATIONS.

-The authors do a nice job highlighting all of the key limitations of the study, but while I appreciate these sober and frank acknowledgements, the overall balance of the discussion seems inappropriate, with long sections dedicated to conceptual speculations that go well beyond the strengths and weaknesses of the data at hand. Upon revision, I'd like to see more balance in the authors' discussion. As written, the paragraph focused on conceptual and methodological limitations undercut most of the claims elaborated in the preceding several paragraphs (e.g., the significance of the findings for anxiety and attention theory).

All of the following points comments relate to this larger critique:

-“Future variations of this task could also utilize control conditions, such as a threat-unrelated cognitive conflicts or non-salient distractors (e.g., changing room temperature), to determine if our results relate to conflict monitoring generally or the attentional demands of anxious arousal specifically.”

-Given the absence of a control condition for distraction, I encourage the authors to temper the urge to draw strong inferences about anxiety. Tasks effects could just reflect distraction, and it would be great to see the authors grapple a bit more with this. How do you ensure attention to the

task? What if there were a condition in which instead of risk of shock, subjects have the chance to win money for each accurately retrieved face?

-The authors find that the ACC and PCC are sensitive to face encoding under threat-of-shock, but there is no non-face condition in their task design. What if this cingulate activation is not reflective of face encoding-related processes, but rather something else (e.g., complex color the image, the shape, or some other feature)?

-“Therefore, we propose a potential extension to the influential attentional control theory (Eysenck et al., 2007), which posits that anxiety reduces goal-directed attentional processing.”

-The task is not really manipulating goal-directed attentional control - there is no competition for attention, as participants are not trying to detect faces in a field of other stimuli or needing to actively ignore other stimuli. The authors might consider softening their inferences to better align with the limitations of the task/sample.

6. FIGURES.

- authors should add a T-bar to Figure 2 to make the brain image interpretable to readers

7. TITLE.

- Clearly describe the article?

ok

8. ABSTRACT.

-Largely ok, but see concerns re: overselling conclusions given the lack of clear control condition (see comments on Discussion)

9. DETAILED ASSESSMENT OF METHOD & ANALYTIC STRATEGY / REPORTING

- If measures or observations are aggregated, do the authors report relevant psychometrics (e.g. alpha, ICC; specify which flavor of ICC).

no they do not

- Do the data meet the assumptions of the specific statistical test (e.g. normality, equal variances)?

not clear as written

- If outlier tests were conducted, are the decision/management rules (e.g., deletion, Winsorization, transformation) adequately motivated and described?

ok

- For each result reported in the text, tables, or figures, are the following clear?: The test coefficient, N, p (1- or 2-sided), df, any descriptive statistics, clearly defined error bars as applicable. Are there adequate descriptive statistics?

See major & minor comments regarding figures

- Are statistical tests justified (statistically and in terms of the aims) and clearly defined for every reported result? If applicable, are nuisance/control variates clearly articulated? Is there an adequate explanation of any control variables that were included/excluded and why, how they influenced variables of interest, and relevant psychometrics?

See concerns re: use of ANOVA framework

- Appropriate adjustments for multiple comparisons?

ok

- If a model was fit, is there evidence of cross-validation? Are the fit statistics or other measures of performance likely to be inflated? Does this need to be acknowledged as a limitation?

ok

- Is incremental validity assessed? Does it need to be?

ok

- If applicable, is any custom software or scripts clearly described? If computer code was used to generate results that are central to the paper's conclusions, do the authors include a statement to indicate whether and how the code can be accessed (include version information as necessary and any restrictions on availability).

ok

- If the authors report and interpret standardized effect sizes, is their interpretation sensible/thoughtful or superficial? Do they adequately distinguish between statistical and practical significance, either in the results or in their discussion of the results? Do they over-interpret the results, given the nature of the sample, paradigm, or model fitting procedures (train/test on the same sample)?

Some concerns about over-interpreting the clinical relevance of these findings (e.g., biomarker language)

- Are the effects substantially **stronger** than one would plausibly expect ('too good to be true'), given the nature of the study? If so, is this adequately addressed in the paper?

ok

- Do the authors adequately distinguish between new discoveries in need of independent replication and theoretical tests? Does the paper need to be revised to address this issue?

ok

- Are the contributions of individual authors adequately described?

ok

For Brain Imaging Papers

- Is the number of blocks, trials or experimental units per session and/ or subjects specified?

ok

- Is the length of each trial and interval between trials specified?

ok

- Is a blocked, event-related, or mixed design being used? If applicable, please specify the block length or how the event-related or mixed design was optimized.

-See major comments for analytic strategy

- For data acquisition, is a whole brain scan used? If not, state area of acquisition and rationale.

ok

- Is the field strength (in Tesla), pulse sequence type (gradient/spin echo, EPI/spiral), field-of-view, matrix size, slice thickness, and TE/TR/ flip angle clearly stated?

ok

- Are the software and specific parameters (model/functions, smoothing kernel size if applicable, etc.) used for data processing and pre-processing clearly stated?

ok

- Is the coordinate space for the anatomical/functional imaging data clearly defined as subject/native space or standardized stereotaxic space, e.g., original Talairach, MNI305, ICBM152, etc?

ok

- If there was data normalization/standardization to a specific space template, is the type of transformation (linear vs. nonlinear) used and image types being transformed clearly described?

ok

- How were anatomical locations determined, e.g., via an automated labeling algorithm (AAL), standardized coordinate database (Talairach daemon), probabilistic atlases, etc.?

ok

- Are the results or threshold based on an ROI (region of interest) analysis? If so, is the rationale clearly described? How were the ROI's defined (functional vs anatomical localization)?

-Yes but there is room for greater consistency across neuroimaging analyses

MINOR / SPECIFIC

- the authors could consider tightening up Figure 2 by reducing white space

-“To determine the spatial specificity of this effect, we used the ACC activation from Garibbo et. al as a region of interest (ROI) alongside a whole-brain analysis, since cingulate regions may form part of a wider neural network involved in anxiety processing (Chavanne & Robinson, 2021).”

-As written, there is no logical connection between using the ACC ROI from the previous paper and determining spatial specificity. What do you mean?

-“Before data analysis, the first four volumes were discarded to allow the magnetic field to stabilize.”

-this sentence should be moved to earlier in the paragraph

-“The first 4 dummy volumes were discarded to allow the magnetic field to stabilize.”

-this sentence is redundant with the later sentence “Before data analysis, the first four volumes were discarded to allow the magnetic field to stabilize.” One of these can be dropped.

-“To make inferences about the neural activation underlying the threat-of-shock potentiated face recognition task...”

-is “potentiated” the most appropriate term here? “Attenuated” may be more appropriate based on directionality of the effect

-The authors may want to consider the differences between incidental vs. integral emotion. For instance, imagine that you are encoding faces while being threatened with shock, as in the present work. How might this differ from more task-relevant anxiety, such as when the faces being encoded are the threat?

-For further details: Shackman, A. J., Sarinopoulos, I., Maxwell, J. S., Pizzagalli, D. A., Lavric, A., & Davidson, R. J. (2006). Anxiety selectively disrupts visuospatial working memory. *Emotion*, 6(1), 40.

-Consider including basic demographic data for participants, both within and across samples (mean age? Percent female?)

-as I commented on with respect to the introduction, what type of biomarker this work would contribute to is unclear in this context. Additionally, if the biomarker identifies face recognition deficits on the basis of brain activation in a threat-of-shock task, it is clear that there is any interest among clinicians or scientists in developing such a biomarker. It also merits comments that it is not guaranteed that a biomarker developed in the present sample would have any relevance to patients with anxiety disorders.

--"For the within-subject modeling we constructed general linear models (GLMs). With the regressors of interest we accounted for both preceding and subsequent state, threat-of-shock (TH) or safety (SF), in our contrasts of encoding and retrieval by including the face stimuli onset times for all combinations of state at encoding (encoding face onsets: THTH, THSF, SFTH, SF SF) and retrieval (retrieval face onsets: THTH, THSF, SFTH, SF SF). Then we also included the contrasts for TH encoding > SF encoding and TH retrieval > SF retrieval. These were convolved with the hemodynamic response function (approximated by a gamma function) using '3dDeconvolve' in AFNI. As nuisance regressors in all within-subject GLMs we included movement-correction parameters (pitch, roll, yaw, z, y, z and derivatives of each motion type) as well as a parameter controlling for the time of shock delivery and presentation time of the safe/threat warnings."

-Did you model it as a block or as a series of events (e.g., one stick per face, but it's basically equivalent to a block since the events are so close together)? It is not entirely clear as written.

-the authors should consider controlling for study/pipeline since they seem to be confounded across samples

EDITORIAL POLICIES

EDITORIAL POLICIES

We ask that you ensure your manuscript complies with our editorial policies and reporting requirements.

To that end, we require revised manuscripts to be accompanied by two completed items: a reporting summary that collects information on study design and procedure, and an editorial policy checklist that verifies compliance with all required editorial policies

Nature Research Reporting Summary

Editorial Policy Checklist

All points on the policy checklist must be addressed. Your revised manuscript can only be sent back to the referees if these checklists are completed and uploaded with the revision.

Notes: If you have submitted a Stage 1 Registered Report, Review, Primer, Comment, or Perspective you do not need to submit these forms. If you have already submitted these forms, you may disregard this request.

* TRANSPARENT PEER REVIEW: Communications Psychology uses a transparent peer review system. This means that we publish the editorial decision letters including Reviewers' comments to the authors and the author rebuttal letters online as a supplementary peer review file. However, on author request, confidential information and data can be removed from the published reviewer reports and rebuttal letters prior to publication. If your manuscript has been previously reviewed at another journal, those Reviewers' comments would not form part of the published peer review file.

REVIEWER EXPERTISE:

Reviewer #1 anxiety, fMRI, computation modeling

Reviewer #2 affective neuroscience, fMRI, computation modeling

REVIEWER REPORTS:

Reviewer #1 (Remarks to the Author):

The study investigated whether an induced anxious state in healthy individuals during encoding and retrieval would lead to impairments in emotional face recognition and the corresponding neural correlates using fMRI. Specifically, the study aimed to evaluate the replicability of previous findings by using a new sample, as well as conducting a mega- and meta-analysis combining three independent samples. The results showed relatively strong evidence supporting the idea that an anxious state during encoding impaired subsequent face recognition. Partially consistent with earlier findings, these impairments were shown to be related to activity in the cingulate cortex. I find this study to be well-designed and to demonstrate an efficient combination of existing data and new data to address an important question in affective science and cognitive science. The findings from this study enhance our knowledge about the neural mechanisms underlying the influence of anxiety on cognitive functioning and have potential implications for anxiety research. The threat-of-shock task used to induce an anxious state is appropriate, and the data analyses are well-suited for addressing the target questions. Below, I have a few major and minor comments, which I believe can be addressed through revision.

Major:

1. My major concern is about the clarity in the theoretical conceptualization. Specifically, in the introduction and discussion, the authors seem to imply that attenuated face processing during an anxious state is related to ACC activity, and ACC is known to be related to attention control. Therefore, the attenuated face processing is related to reduced attention allocation during an anxious state. Further, the authors seem to imply that the task evoked anxious apprehension, which is the cognitive dimension of anxiety, but the attention demand is because of the anxious arousal, which is the physiological dimension of anxiety. If this is what the authors meant to communicate, then I would suggest making the reasoning more explicit. However, it remains unclear how anxious

apprehension, the cognitive dimension of anxiety, leads to the attention allocation difficulty related to anxious arousal, the physiological dimension.

The reviewer is proposing a distinction between the cognitive and physiological aspects of anxiety, which is an interesting one, but in practice the anxious state involves a range of changes that occur across the whole brain and body. The present work is agnostic to this conceptual distinction and simply relies on the literature showing that the threat-of-shock anxiety manipulation promotes cognitive and physiological changes, which includes anxious arousal. Specifically, we measure behavioural and BOLD changes to identify the replicable impacts of threat of unpredictable shock on these changes. Therefore, our task and measures are unfortunately not designed to distinguish between different dimensions of anxiety and tease apart whether or how they might differentially affect attention allocation.

We clarify this now and provide references from our previous reviews of this literature in the introduction:

“The threat-of-shock manipulation reliably induces state anxiety through the anticipation of unpredictable non-painful electrical shocks (A. Schmitz & Grillon, 2012) This promotes a range of physiological, neural and cognitive changes similar to those found in pathological anxiety (Chavanne & Robinson, 2021; Robinson et al., 2013) and has been proposed as an intermediate bridge between basic and clinical research (Grillon et al., 2019).” – Main Manuscript, page 2-3

In response to this and other helpful comments (such as the one below) we have made attempts to improve the clarity of our theoretical conceptualization, especially when relating attenuated face processing and ACC activity to attentional conflict as a potential mechanism, by referring more clearly to the theoretical accounts this suggestion builds on:

“This hypothesized effect aligns with theoretical accounts that suggest state anxiety may compete for top-down allocation of attentional resources, which could be signalled by the increased ACC activation, and that face identity encoding may be particularly vulnerable to this disruption (Eysenck et al., 2007; Palermo & Rhodes, 2007; Pessoa, 2009).” – Main Manuscript, page 4

Similarly, in the discussion, we attempted to improve our conceptualization by clearly referring to the theoretical accounts supporting our interpretation of the observed effects:

“Consistent with proposals of selective attention, more attentional resources may be allocated to task-unrelated threat-signals during perceptual processing, thus reducing attention allocated to face encoding (Bishop, 2008; Lavie, 2005; Lavie et al., 2004; Palermo & Rhodes, 2007). In line with other work on dual process competition it is specifically in the case of task-irrelevant anxiety, which we were interested in eliciting by our threat-of-shock paradigm, that arousal signals compete with

cognitive processes, so this may be unaffected or reversed when the source of threat is task-relevant (Lago et al., 2022; Robinson et al., 2013; Shackman et al., 2006).” – Main Manuscript, page 17

2. Relatedly, the theories driving the original hypotheses are unclear to me. I hope the authors can further elaborate and/or clarify the theoretical advancements gained from these findings. In addition, the authors may add implications on emotional control.

We thank the reviewer for asking us to elaborate on this and have rewritten this section of the introduction accordingly:

“Our primary analysis pertained to replicating the behavioural reduction in recognition accuracy for faces previously encoded during a threat-of-shock anxious state, and associated increase in brain activation in the ACC during the anxious face encoding stage. This hypothesized effect aligns with theoretical accounts that suggest state anxiety may compete for top-down allocation of attentional resources, which could be signalled by the increased ACC activation, and that face identity encoding may be particularly vulnerable to this disruption (Eysenck et al., 2007; Palermo & Rhodes, 2007; Pessoa, 2009). In this replication study, we sought to clarify if this behavioural effect emerges reliably in a large independent sample and if the underlying neural activity is specific to the ACC. We therefore used the ACC activation from Garibbo et. al as a region of interest (ROI) alongside a whole-brain analysis, since cingulate regions form part of a wider neural network involved in attentional and anxiety processing (Chavanne & Robinson, 2021). In our secondary neural analysis, we further hypothesized to replicate an increase in activation in the bilateral hippocampus when the state, threat-of-shock or safety, during face retrieval is congruent with the state during prior encoding (ROI from Garibbo et al., 2019). This aligns with extensive evidence implicating the hippocampus in the reactivation of context-item associations formed during encoding (Eichenbaum et al., 2012), but remains to be replicated for this paradigm.” – Main Manuscript, pages 3-4

3. While the authors claimed that part of the motivation to conduct this study is the clinical relevance, specifically, it is important to determine the replicability of neuroimaging and behavioral effects in the general population before investigating the individual differences related to clinical conditions. I highly agree with the authors on this point of establishing within-subject effects in the general population first. But I encourage the authors to point out that even a robust within-subject effect does not warrant individual differences.

We are glad the reviewer agrees with the value and motivation of this replication study and completely support the notion that a within-subject effect should not be equated with individual differences related to clinical conditions, but merely as a foundation to warrant further investigation of these differences. To clarify that we have changed our formulation in the introduction as follows:

“Given the increasing interest in anxiety disorder biomarkers and clinical relevance of anxiety-induction research, it is critical to determine the replicability of the neuroimaging and behavioural effects of within-subject state anxiety manipulations in non-symptomatic individuals before investigating how these differ in clinical groups.” – Main Manuscript, page 3

4. Regarding the task, the face stimuli included emotional faces. However, it is unclear what the rationale is for including emotionality in this task in the present paper.

Thank you for this comment, we are happy to clarify.

The reason we have included emotionality in this task is because emotional face stimuli are more salient than neutral stimuli and their processing is particularly relevant and affected by anxiety, both in clinical groups and under induced anxiety stated in the general population.

Thanks to the reviewer’s comment we have now clarified in the introduction that the effect of interest and references we provided pertain to emotional face stimuli:

*“anxiety has been associated with abnormalities in the recognition of previously-seen **emotional face stimuli** (Attwood et al., 2013; Bolton & Robinson, 2017; Garibbo et al., 2019; Jarros et al., 2012; Surcinelli et al., 2006; Tindall et al., 2021).” – Main Manuscript, page 2*

We have also reformulated the following sentence in the introduction to make clear, as the reviewer kindly pointed out, the importance and rationale for the focus on emotional face stimuli:

“This function is highly relevant since our environments are predominately social and the ability to recognize familiar faces is pertinent to navigating them successfully. It is also known that emotional faces are particularly salient stimuli, which benefit from enhanced processing over non-salient stimuli by preferentially recruiting attentional resources (Compton, 2003; Palermo & Rhodes, 2007), and are prone to anxiety-related alterations (Chen et al., 2002; Edwards, 2014; Fox et al., 2005; Kavcıoğlu et al., 2021).” – Main Manuscript, page 2

5. fMRI methods. Please clarify if this task is a single scan session task or multiple sessions, with each corresponding to a block.

We apologize if this was unclear and confirm that this task is a single scan session task rather than each block spread across multiple sessions. To avoid this confusion, we have added the following (in bold) to the Participants subsection of the Methods section:

“We collected data for a total sample of n=98 participants aged between 18-64, who completed the threat-of-shock potentiated face recognition task during a single fMRI scanning session between December 2017 and May 2022.” – Main Manuscript, page 5

6. The discussion highlighted the consistency between the current PCC and the previous ACC findings. The authors could further elaborate on the potential differential roles. This may have different theoretical implications.

We thank the reviewer for this suggestion and added the following to our discussion to elaborate on the differential roles of ACC and PCC in the literature:

“While increased activation in the ACC has been reported across a range of task-based and anxiety-potentiated paradigms (Bush et al., 2000; Etkin et al., 2011; Maier et al., 2012; Straube et al., 2009), the PCC is more commonly associated with the default mode network and introspective resting states (Greicius et al., 2003; Leech et al., 2011, 2012), and its involvement in this task may have only emerged due to the unusually large sample size.” – Main Manuscript, page 18

Minor:

7. In the introduction, the authors may consider citing APA and RDoC works when defining anxiety.

We thank the reviewer for the suggestion to provide a citation when defining anxiety and now made this adjustment in the introduction by referencing the National Institute of Mental Health Strategic Plan, which unveiled the RDoC:

“The anxious state, characterized by an aversive anticipation of potential but unpredictable threats, involves increased vigilance and neurophysiological arousal (see Research Domain Criteria in The National Institute of Mental Health Strategic Plan, 2008) as well as alterations in neurocognitive processing (Chavanne & Robinson, 2021).” – Main Manuscript, page 2

8. Also in the first paragraph, references seem to be needed when the authors wrote that inconsistent findings exist across samples and different paradigms.

We are glad the reviewer pointed out that that this section of the introduction would benefit from references and have amended it accordingly:

“However, empirical findings on the precise causal effects of state anxiety on face recognition are sparse and inconsistent in the manipulations, experimental paradigms and encoding versus retrieval stage of face memory processing investigated across different samples (Attwood et al., 2013, 2015; Curtis et al., 2015; Moon et al., 2016; Tindall et al., 2021).” – Main Manuscript, page 2

9. In the methods section, it would be helpful to briefly describe the samples of the prior studies in the present article for ease of reading.

We thank the reviewer for this suggestion to improve the clarity and ease of reading of our methods section and added the following to the Participants subsection:

“For additional meta-analytic analyses, we also collated data from two studies. These included n=86 participants who had completed the threat-of-shock potentiated face recognition task during a behavioural testing session (see Bolton & Robinson, 2017) and n=32 participants who had completed it during a fMRI scanning session (see Garibbo et al., 2019).” – Main Manuscript, page 14

10. The Meta- and Mega-analysis methods section seems to be under-referenced.

We thank the reviewer for this comment and have added two relevant references for the meta- and mega-analysis methods. Note that following other reviewer suggestions, we have moved the meta-analysis methods and results to the supplementary materials and only present the mega-analysis methods in the main manuscript. We added the following references to the methods section:

“We also performed a mega-analysis, considered the ‘gold-standard’ of meta-analytic approaches and also referred to as individual-participant data (IPD) meta-analysis (Eisenhauer, 2021; Tierney et al., 2015).” – Main Manuscript, page 17

Eisenhauer, J. G. (2021). Meta-analysis and mega-analysis: A simple introduction. Teaching Statistics, 43(1), 21–27. <https://doi.org/10.1111/test.12242>

Tierney, J. F., Vale, C., Riley, R., Smith, C. T., Stewart, L., Clarke, M., & Rovers, M. (2015). Individual Participant Data (IPD) Meta-analyses of Randomised Controlled Trials: Guidance on Their Use. PLOS Medicine, 12(7), e1001855. <https://doi.org/10.1371/journal.pmed.1001855>

Reviewer #2 (Remarks to the Author):

#####

Investigating the replicability of neural mechanisms underlying anxiety-attenuated encoding of emotional faces

Buehler...& Robinson

#####

The present study leverages a relatively large sample to replicate and extend prior work on the impacts of threat-of-shock-induced anxiety on the encoding and recognition stage of emotional face processing. The authors conclude that anxiety impairs face recognition, possibly due to attentional interference during encoding, and that the cingulate cortex is critical to this process.

Key Strengths (Significance, Innovation, Rigor, Approach):

large pooled sample

attempt to grapple with the replication crisis

pre-registered approach

clear articulation of the gap that the present work seeks to address

thoughtful acknowledgement of key limitations in discussion

Key Weaknesses (Significance, Innovation, Rigor, Approach):

as written, the rationale for the inclusion of both the meta- and mega-analyses is undermotivated in the introduction

relatedly, the authors do not have a clear, consistent approach for the analyses across studies/samples...authors might consider either doing replication analyses for behavior only or just doing the same thing for behavior and for brain metrics

need for a more balanced discussion (not overselling clinical relevance/use of biomarkers/limitations based on sample composition and study design)

In the section that follows, I provide a few suggestions for further strengthening the manuscript.

MAJOR / GENERAL

INTRODUCTION.

11. As written, the rationale for the inclusion of both the meta- and mega-analyses is undermotivated in the introduction. While both mega- and meta- analytic approaches are an important step forward, it is unclear whether both are necessary in the present context. I encourage the authors to consider the complimentary strengths and weaknesses of including both. Perhaps they are sufficiently redundant so as to scrub the meta-analysis.

We are grateful to the reviewer for this thoughtful suggestion. We were initially motivated to report in our main manuscript the complementary results of both a standard meta-analysis as well as a less commonly reported but considered 'gold-standard' mega-analysis approach. With the former we hoped to demonstrate what effects are consistent across the studies, when accounting for their different effect sizes and weighting them by their inverse sampling variance, while the latter approach allowed us to take advantage of the identical task paradigm and unique access to raw data to replicate our analyses and determine what effects are robust in a combined sample. However, we do agree with the reviewer that in the present context there is sufficient redundancy to feature only the mega-analysis in the main manuscript. We also see the benefit of consistency in featuring the same mega-analysis approach used for both the behavioural data as well as the neuroimaging data, which was only collected for two of the studies and is therefore suitable for the mega- but not meta-analysis. We now removed the behavioural meta-analyses from the main manuscript, to the supplementary material.

In the introduction, we accordingly only introduce the mega-analytic approach, as follows:

“Finally, we combined the original behavioural and neural data from Bolton & Robinson (2017) and Garibbo et al. (2019) with our sample to complement our primary and secondary analyses with a mega-analytic approach that indicates what effects are robust across the studies (Eisenhauer, 2021; Tierney et al., 2015).” – Main Manuscript, page 4

From the results section we moved the parts pertaining to the meta-analysis from the main manuscript to the supplementary materials and adjusted the following subsection in the methods section of the main manuscript accordingly, to only feature the mega-analysis:

“Exploratory Mega-Analyses of Behaviour across studies

We also performed a mega-analysis, considered the ‘gold-standard’ of meta-analytic approaches and also referred to as individual-participant data (IPD) meta-analysis (Eisenhauer, 2021; Tierney et al., 2015). For this we combined the raw behavioural data from all participants (n=210), across the current study (referred to as Buehler et al. 2023: n=92) and the two previous studies utilizing the same threat-of-shock potentiated face recognition task (Bolton & Robinson, 2017: n=86; Garibbo et al., 2019: n=32). Based on the aforementioned behavioural hypotheses, this was done to assess the robustness of a) the effect of state (threat-of-shock or safety) distinctly at encoding and retrieval, as well as b) the effect of state congruency at retrieval on face recognition accuracy across all three studies. We therefore used the same ANOVA models as in the primary and secondary analyses but accounting for study as a between-subjects factor. Of interest were the main effects of encoding state and retrieval state on recognition accuracy.” – Main Manuscript, page 17-18

The following description of the meta-analysis, alongside its results, can now be found in the supplementary material:

“Exploratory Summary Statistic Meta-Analysis of Behaviour across studies

*In addition to the mega-analysis, we report the results of a standard meta-analysis, which utilizes the summary statistics and a weighting mechanism to account for differences in precision between different samples (Eisenhauer, 2021). We utilized the multivariate random effects meta-analysis model fit using the default `rma.mv` function from the `metafor` package in R (Viechtbauer, 2010) with the following syntax in R: `rma.mv(yi=d, V= Var(d), slab=study, data, random=~1|study)`. In this model the individual study estimates were weighted using the inverse sampling variance of the study relative to the estimated amount of heterogeneity across all studies. Due to the within-subjects design, we calculated the Cohen’s *d* effect size (*d*) and its variance (*Var(d)*) from the ANOVA *F*-statistics and sample sizes as follows: $d = \sqrt{F/N}$, $Var(d) = 1/N + d^2/2N$.” – Supplementary Material, page 2*

12. “Given the clinical relevance of this research, such as the potential development of anxiety-disorder biomarkers, it is critical to first determine the replicability of neuroimaging and behavioural effects pertaining to induced anxiety-cognition interactions in the general population. For instance, if unrelated brain areas stochastically emerge for the same contrast across different healthy samples, they would have no value as pre-clinical biomarkers. Instead, substantially larger sample sizes or meta-analyses may be required to ascertain the stability of evoked activations.”

-Some of the aspirational clinical significance noted in the introduction seems far-fetched given that this is a super healthy (asymptomatic with no lifetime family history) sample using a threat-of-shock task. This isn't a population representative sample.

-Additionally, I encourage the authors to consider minimizing or eliminating the biomarker content, as it deters from the real strengths of the paper (e.g., large N, pre-registered approach). It is not clear what type of biomarker this work would yield (e.g., case-control biomarker, one relevant for treatment, or one for identifying risk?), and it is further unclear whether a biomarker derived from this healthy sample would prove useful.

We thank the reviewers for this feedback and agree with these suggestions for improving the conceptual clarity in our introduction. Please see the relevant section of the introduction below, where we now refer to our sample explicitly as non-symptomatic rather than suggesting it may be representative of the general population. We have also attempted to make more clear that the clinical relevance and implications for biomarker development this study provides merely relates to the importance of establishing the replicability of behavioural and neural effects related to state anxiety in non-clinical samples.

“Given the increasing interest in anxiety disorder biomarkers and clinical relevance of anxiety-induction research, it is critical to determine the replicability of the neuroimaging and behavioural effects of within-subject state anxiety manipulations in non-symptomatic individuals before investigating how these differ in clinical groups.” – Main Manuscript, page 3

We also removed the biomarker content from the discussion as suggested and now state our implications as follows:

“This has important implications for our understanding of anxiety disorders and their treatment response. Specifically, it is crucial to determine the replicability of behavioural and neuroimaging effects stemming from state anxiety alterations in non-clinical samples, to help ensure future comparisons with patient populations reflect commonalities and differences that are clinically meaningful rather than driven by random variability from unreliable tasks that fail to replicate (Chavanne & Robinson, 2021).” – Main Manuscript, page 12

METHOD.

13. As noted above, some of the aspirational clinical significance and biomarker language noted in the introduction seems far-fetched given that this is a super healthy (asymptomatic with no lifetime family history) sample using a threat-of-shock task. This isn't a population representative sample. Careful re-consideration of this language will allow the genuine strengths of the paper to stand out.

We are grateful for this feedback and refrain from aspirational clinical significance and biomarker language, please see the detailed changes described in our previous responses.

ANALYTIC STRATEGY.

RESULTS.

14. The paper would benefit from clearly distinguishing hypothesis testing from exploratory testing. For example, in the introduction, the authors hypothesize that “anxious encoding would be associated at the neural level with increased activation in the ACC.” However, in the data analysis section of the method, they write, “[w]e used the resampled group-level dorsal ACC activation map (using AFNI’s 3dmaskave) from Garibbo et al., (2019) as an ROI for the following contrast of interest: threat-of-shock > safety, during encoding and retrieval separately.” It is unclear to the reader why retrieval is being examined when the stated hypothesis pertains to encoding.

We thank the reviewer for pointing this out and apologize for not making clear the rationale for also examining the retrieval stage, which is to investigate if our hypothesized threat-of-shock effects are specific to the encoding stage rather than also occurring for retrieval. We have rephrased this section in the introduction to make this clear:

*“Our primary analyses pertained to replicating the behavioural reduction in recognition accuracy for faces previously encoded during a threat-of-shock anxious state, and associated increase in brain activation in the ACC during the anxious face encoding **but not retrieval** stage.” – Main Manuscript, page 3-4*

We added this adjustment to the relevant methods sections:

“Primary Behavioural Hypothesis: Reduced recognition of faces encoded under threat-of-shock compared to safety.

To address our primary behavioural hypothesis that threat-of-shock at encoding but not retrieval impairs face recognition accuracy, we statistically tested for a main effect of state (threat-of-shock or safety) at the encoding and retrieval stages, as well as their interaction on face recognition accuracy using a within-subjects factorial analysis of variance (ANOVA). – Main Manuscript, page 17

Primary Neural Hypothesis: Increased ACC activation while encoding faces under threat-of-shock compared to safety.

To address our primary neural hypothesis that threat-of-shock at encoding but not retrieval is associated with increased ACC activation, we used the resampled group-level dorsal ACC activation

map (using AFNI's 3dmaskave) from Garibbo et al., (2019) as an ROI for the following contrast of interest: threat-of-shock blocks > safety blocks, during encoding and retrieval separately.” – Main Manuscript, page 19-20

In the same manner, we added this adjustment to the relevant results section:

“To determine if threat-of-shock at encoding but not retrieval impairs face recognition in our sample, we tested for a main effect of state (threat-of-shock or safety) at the encoding and retrieval stage, as well as their interaction on face recognition accuracy using a within-subjects factorial analysis of variance (ANOVA)” – Main Manuscript, page 5

15. Moreover, a consistent use of nomenclature would aid the reader in following along. THTH versus TofS becomes somewhat difficult to keep straight. Why not just consistently use Threat vs Safety?

We thank the reviewer for this comment and have made the nomenclature consistent by referring to the conditions as threat-of-shock or safe and removing all the other THTH, ToS, etc. abbreviations, to make it more intuitive for the reader. See for instance the following changes in the description of the task:

“This experimental task was identical to that in Garibbo et al. (2019) (which was in turn replicated from Bolton and Robinson (2017)) and completed in an fMRI scanner. The task consisted of 4 blocks, each involving an encoding phase followed by a retrieval phase, alternating between a threat-of-shock state and safe control state (see figure 1a and 1b). This constituted a 2 (safe vs threat-of-shock) by 2 (encoding vs retrieval) design and resulted in the following combinations of encoding and retrieval respectively: safe followed by threat-of-shock, safe followed by safe, threat-of-shock followed by safe, threat-of-shock followed by threat-of-shock.”.- Main Manuscript, page 14-15

16. Another way to compare threat and safe would be to compare TS and TT to SS and ST...the primary behavioral hypothesis is framed in ANOVA language, but as written, it is not clear that ANOVA is the most appropriate overarching analytic framework. Further clarity around how contrasts were set up would greatly aid the reader.

We thank the reviewer for this feedback and apologize that this was unclear in the original manuscript. Indeed, our analyses compare TS and TT to SS and ST. In the Methods Behavioural Data Analysis subsection for the primary hypothesis we now clarified this and the use of the ANOVA framework:

“Specifically, the ANOVA framework allowed us to contrast face recognition accuracy for all state combinations of threat-of-shock at encoding (i.e., followed by either threat-of-shock or safety at

retrieval) with safety at encoding (i.e., followed by either threat-of-shock or safety at retrieval), and for all combination of state at retrieval in the same manner.” – Main Manuscript, page 17

17. Likewise, for congruence is it TT and SS vs. ST and TS (i.e., congruent vs incongruent)? Stating the contrasts more explicitly would enhance the report.

We confirm that this is correct and have now clarified these contrasts in the Methods Behavioural Data Analysis subsection:

“Specifically, the ANOVA framework allowed us to contrast face recognition accuracy at retrieval for all congruent state combinations (threat-of-shock encoding followed by threat-of-shock retrieval, safety encoding followed by safety retrieval) with all incongruent state combinations (threat-of-shock encoding followed by safety retrieval, safety encoding followed by threat-of-shock retrieval).” – Main Manuscript, page 17

18. Put differently, the authors might consider adopting theory-driven focused contrasts (e.g., encode under threat/retrieve under threat and encode under safety/retrieve under safety compared to encode under safety/retrieve under threat and encode under threat/retrieve under safety). This may be clearer than using ANOVA main effect language since the results are not really main effects in the traditional way.

We hope our responses to the previous comments provide the necessary clarification and demonstrate that our results are based on the most comprehensive ad theory-driven combination of contrasts that are suitable for our task design.

DISCUSSION / IMPLICATIONS.

The authors do a nice job highlighting all of the key limitations of the study, but while I appreciate these sober and frank acknowledgements, the overall balance of the discussion seems inappropriate, with long sections dedicated to conceptual speculations that go well beyond the strengths and weaknesses of the data at hand. Upon revision, I'd like to see more balance in the authors' discussion. As written, the paragraph focused on conceptual and methodological limitations undercut most of the claims elaborated in the preceding several paragraphs (e.g., the significance of the findings for anxiety and attention theory).

All of the following points comments relate to this larger critique:

19. “Future variations of this task could also utilize control conditions, such as a threat-unrelated cognitive conflicts or non-salient distractors (e.g., changing room temperature), to determine if our results relate to conflict monitoring generally or the attentional demands of anxious arousal specifically.”

We have now removed this section to reduce the elaborate discussion of limitations as suggested and, following the next reviewer comment, rewritten as follows:

“Future variations of this task could also utilize further control conditions, such as threat-unrelated distractors or the chance to win money for accurately retrieved faces.” – Main Manuscript, page 13

20. Given the absence of a control condition for distraction, I encourage the authors to temper the urge to draw strong inferences about anxiety. Tasks effects could just reflect distraction, and it would be great to see the authors grapple a bit more with this. How do you ensure attention to the task? What if there were a condition in which instead of risk of shock, subjects have the chance to win money for each accurately retrieved face?

We thank the reviewer for raising a very important consideration when interpreting the results from any experimental design, which is to acknowledge that not all conditions (e.g., distraction) can be controlled for with a single task.

While we agree no inferences should be made explicitly about pathological anxiety based on this task, over a decade of work has noted overlap between the cognitive and physiological effects of the threat-of-shock relative to pathological anxiety. We provide the following description and references in the introduction:

“The threat-of-shock manipulation reliably induces state anxiety through the anticipation of unpredictable non-painful electrical shocks (A. Schmitz & Grillon, 2012). This promotes a range of physiological, neural and cognitive changes similar to those found in pathological anxiety (Chavanne & Robinson, 2021; Robinson et al., 2013) and has been proposed as an intermediate bridge between basic and clinical research (Grillon et al., 2019).” – Main Manuscript, page 2-3

We certainly agree that the distraction resulting from the threat-of-shock manipulation may have shifted attention away from the encoding of external face stimuli to make this more clear, we have rewritten the section in the discussion:

“At a mechanistic level, the observed impairment may have occurred as a result of more attentional resources allocated away from the perceptual processing of external face stimuli and towards the highly salient threat-signal. While we ensured our results were not driven by participants who did not attend to the task at all (see behavioural analysis in methods section), it could be that these

attentional lapses occurred on a trial basis throughout the encoding stage.” – Main Manuscript, page 10

We are particularly grateful for the reviewer’s comment on how to ensure attention to the task, as this is indeed a challenge in many psychological experiments and difficult to measure directly without interfering with the task. The way we addressed this in the present study was by ensuring our results are representative of participants that performed above chance level across all conditions on average, which would not be the case if they do not pay any attention to the task. We did this by rerunning our behavioural analyses without participants that performed below chance level (face recognition accuracy below 0.5) in all of the four conditions. However, this was only the case for 4 of our 92 participants and did not change the effects we observed, so we failed to mention this in the initial manuscript. We have now we added the following to the Behavioural Data Analysis subsection of the Methods:

“To ensure our results were not driven by participants who may have failed to understand or attend to the task, we reran our analyses excluding participants whose behavioural performance on the task was at or below chance level (mean recognition accuracy of <0.5) across all four conditions. However, this only applied to 4 participants and did not change the significance of any effects.” – Main Manuscript, page 17

Finally, we agree that how a different task paradigm with a chance to win money instead of a threat-of-shock condition would affect face recognition is another very interesting empirical question, that we would be intrigued to see investigated. We now note that this would be a valuable control condition in the discussion section:

“Future variations of this task could also utilize other threat-unrelated control conditions, such as the chance to win money for accurately retrieved faces.” – Main Manuscript, page 13

21. The authors find that the ACC and PCC are sensitive to face encoding under threat-of-shock, but there is no non-face condition in their task design. What if this cingulate activation is not reflective of face encoding-related processes, but rather something else (e.g., complex color the image, the shape, or some other feature)?

We completely agree that not controlling for other non-face stimuli is one of the limitations of this task and it could certainly be the case that the ACC and PCC effects relate to other stimuli or aspects of the image such as colour, shape, etc. In other words, while our task clearly shows that activation in the ACC and PCC regions is sensitive to face encoding under threat-of-shock, this by no means rules out other features of these stimuli and we would be delighted to see the specificity of these effects explored in future work.

We would like to clarify that the focus of the present paradigm was limited to emotional face stimuli specifically rather than a comprehensive exploration of the differential effects of different types of stimuli, and we elaborate on the rationale for this in the introduction following other reviewer comments as follows:

“Yet, the precise functions and effects of state anxiety on cognition remain poorly understood. For instance, anxiety has been associated with abnormalities in the recognition of previously-seen emotional face stimuli (Attwood et al., 2013; Bolton & Robinson, 2017; Garibbo et al., 2019; Jarros et al., 2012; Surcinelli et al., 2006; Tindall et al., 2021). This function is highly relevant since our environments are predominately social and the ability to recognize familiar faces is pertinent to navigating them successfully. It is also known that emotional faces are particularly salient stimuli, which benefit from enhanced processing over non-salient stimuli by preferentially recruiting attentional resources (Compton, 2003; Palermo & Rhodes, 2007), and are prone to anxiety-related alterations (Chen et al., 2002; Edwards, 2014; Fox et al., 2005; Kavcıoğlu et al., 2021).” – Main Manuscript, page 2

We also clarify this limitation in the discussion:

“it remains unknown if the behavioural and neural effects we found are specific to faces or would also extend to other emotionally salient and non-salient stimuli. It is plausible, for instance, that effects are driven by features of the stimuli beyond the social and emotional relevance (e.g. the shape, colour or other generic feature).” – Main Manuscript, page 12

22. “Therefore, we propose a potential extension to the influential attentional control theory (Eysenck et al., 2007), which posits that anxiety reduces goal-directed attentional processing.”

The task is not really manipulating goal-directed attentional control - there is no competition for attention, as participants are not trying to detect faces in a field of other stimuli or needing to actively ignore other stimuli. The authors might consider softening their inferences to better align with the limitations of the task/sample.

We agree that a softening of these inferences is appropriate and adjusted the section referred to accordingly:

“In line with the influential attentional control theory, which posits that anxious arousal reduces the attentional processing of external stimuli (Eysenck et al., 2007), there may be an increase in internally directed attention to one’s physiological response under threat-of-shock. This, in turn, could be signalled by increased anterior and posterior cingulate cortex activation in the brain and impair the ability to accurately encode face stimuli in our task.” – Main Manuscript, page 19

FIGURES.

23. authors should add a T-bar to Figure 2 to make the brain image interpretable to readers

We thank the reviewer for pointing this out and have now added a T-bar to figure 2.

24. TITLE. Clearly describe the article? ok

25. ABSTRACT. Largely ok, but see concerns re: overselling conclusions given the lack of clear control condition (see comments on Discussion)

We have now rewritten the abstract as follows:

“Anxiety involves the anticipation of aversive outcomes and can impair neurocognitive processes, such as the ability to recall faces encoded during the anxious state. It is important to precisely delineate and determine the replicability of these effects using causal state anxiety inductions in the general population. This study therefore aimed to replicate prior research on the distinct impacts of threat-of-shock-induced anxiety on the encoding and recognition stage of emotional face processing, in a large asymptomatic sample (n=92). We successfully replicated previous results demonstrating impaired recognition of faces encoded under threat-of-shock. This was supported by a mega-analysis across three independent studies using the same paradigm (n=211). Underlying this, a whole-brain fMRI analysis revealed enhanced activation in the posterior cingulate cortex (PCC), alongside previously seen activity in the anterior cingulate cortex (ACC) when combined in a mega-analysis with the fMRI findings we aimed to replicate. We further found replications of hippocampus activation when the retrieval and encoding states were congruent. Our results support the notion that state anxiety disrupts face recognition, potentially due to attentional demands of anxious arousal competing with affective stimuli processing during encoding and suggest that regions of the cingulate cortex play pivotal roles in this.” – Main Manuscript, page 1

DETAILED ASSESSMENT OF METHOD & ANALYTIC STRATEGY / REPORTING

26. If measures or observations are aggregated, do the authors report relevant psychometrics (e.g. alpha, ICC; specify which flavor of ICC). no they do not

We now report the internal consistency in the supplementary materials:

Since the 36 trials were aggregated to compute the mean face recognition accuracy at retrieval for each of the four task blocks, per subject, we now report Cronbach’s Alpha as a measure of internal consistency for the trial-level data. We report the alpha for each condition separately and confirm that the internal consistency ranges from acceptable (0.7-0.8) to good (0.8-0.9):

1. Block with safety at encoding followed by safety at retrieval: 0.831
2. Block with safety at encoding followed by threat-of-shock at retrieval: 0.795
3. Block with threat-of-shock at encoding followed by safety at retrieval: 0.729
4. Block with threat-of-shock at encoding followed by threat-of-shock at retrieval: 0.791

– *Supplementary Material, page 1*

27. Do the data meet the assumptions of the specific statistical test (e.g. normality, equal variances)?

not clear as written

To make this clear we have now included the following in our main manuscript methods section:

“To ensure that this data met the assumptions of the statistical test (ANOVA) used in our behavioural analysis, we ran Shapiro-Wilk normality tests using the `shapiro_test()` function in R to confirm there is no significant deviation from the normal distribution in our model residuals for the effect of encoding state ($p=0.58$) and retrieval state ($p=0.72$). The `anova_test()` function in R from the `rstatix` package, which we used to compute the ANOVA models, automatically checks the assumption of sphericity internally using the Mauchly’s test.” – Main Manuscript, page 7

28. If outlier tests were conducted, are the decision/management rules (e.g., deletion, Winsorization, transformation) adequately motivated and described?

ok

29. For each result reported in the text, tables, or figures, are the following clear?: The test coefficient, N, p (1- or 2-sided), df, any descriptive statistics, clearly defined error bars as applicable. Are there adequate descriptive statistics?

See major & minor comments regarding figures

See our response to the comment 23 regarding figures.

30. Are statistical tests justified (statistically and in terms of the aims) and clearly defined for every reported result? If applicable, are nuisance/control variates clearly articulated? Is there an adequate explanation of any control variables that were included/excluded and why, how they influenced variables of interest, and relevant psychometrics?

See concerns re: use of ANOVA framework

See our response to the comment 16 regarding the use of ANOVA framework.

“Specifically, the ANOVA framework allowed us to contrast face recognition accuracy for all state combinations of threat-of-shock at encoding (i.e., followed by either threat-of-shock or safety at

retrieval) with safety at encoding (i.e., followed by either threat-of-shock or safety at retrieval), and for all combination of state at retrieval in the same manner.” – Main Manuscript, page 8

31. Appropriate adjustments for multiple comparisons?

ok

32. If a model was fit, is there evidence of cross-validation? Are the fit statistics or other measures of performance likely to be inflated? Does this need to be acknowledged as a limitation?

ok

33. Is incremental validity assessed? Does it need to be?

ok

34. If applicable, is any custom software or scripts clearly described? If computer code was used to generate results that are central to the paper's conclusions, do the authors include a statement to indicate whether and how the code can be accessed (include version information as necessary and any restrictions on availability).

ok

35. If the authors report and interpret standardized effect sizes, is their interpretation sensible/thoughtful or superficial? Do they adequately distinguish between statistical and practical significance, either in the results or in their discussion of the results? Do they over-interpret the results, given the nature of the sample, paradigm, or model fitting procedures (train/test on the same sample)?

36. Some concerns about over-interpreting the clinical relevance of these findings (e.g., biomarker language)

We have removed any strong claims of biomarker or clinical relevance (see response to reviewer comment 3, 12, 25)

37. Are the effects substantially **stronger** than one would plausibly expect ('too good to be true'), given the nature of the study? If so, is this adequately addressed in the paper?

ok

38. Do the authors adequately distinguish between new discoveries in need of independent replication and theoretical tests? Does the paper need to be revised to address this issue?

ok

39. Are the contributions of individual authors adequately described?

ok

For Brain Imaging Papers

40. Is the number of blocks, trials or experimental units per session and/ or subjects specified?

ok

41. Is the length of each trial and interval between trials specified?

ok

42. Is a blocked, event-related, or mixed design being used? If applicable, please specify the block length or how the event-related or mixed design was optimized.

See major comments for analytic strategy

See our previous response to the comments 14-17 regarding analytic strategy.

43. For data acquisition, is a whole brain scan used? If not, state area of acquisition and rationale.

ok

44. Is the field strength (in Tesla), pulse sequence type (gradient/spin echo, EPI/spiral), field-of-view, matrix size, slice thickness, and TE/TR/ flip angle clearly stated?

ok

45. Are the software and specific parameters (model/functions, smoothing kernel size if applicable, etc.) used for data processing and pre-processing clearly stated?

ok

46. Is the coordinate space for the anatomical/functional imaging data clearly defined as subject/native space or standardized stereotaxic space, e.g., original Talairach, MNI305, ICBM152, etc?

ok

47. If there was data normalization/standardization to a specific space template, is the type of transformation (linear vs. nonlinear) used and image types being transformed clearly described?

ok

48. How were anatomical locations determined, e.g., via an automated labeling algorithm (AAL), standardized coordinate database (Talairach daemon), probabilistic atlases, etc.?

ok

49. Are the results or threshold based on an ROI (region of interest) analysis? If so, is the rationale clearly described? How were the ROI's defined (functional vs anatomical localization)?

Yes but there is room for greater consistency across neuroimaging analyses

We have improved the consistency across our neuroimaging analyses in response to this and other helpful reviewer comments. In particular, we now ensure consistency of the neuroimaging analyses for all fMRI ROI, whole-brain and mega-analyses models and matched these with the behavioural analyses by using ANOVA models. We included the same control for study as a between-subjects factor in our fMRI whole-brain and ROI mega-analysis to make this consistent with our behavioural analysis. We note that the inference has remained the same, but we have updated the methods and results section accordingly:

See our changes below:

Methods:

“To address our primary neural hypothesis that threat-of-shock at face encoding but not retrieval is associated with increased ACC activation, we used the resampled group-level dorsal ACC activation map (using AFNI’s 3dmaskave) from Garibbo et al., (2019) as an ROI for the following contrast of interest: faces during threat-of-shock blocks > safety blocks, at encoding and retrieval separately. To ensure consistency with the behavioural analysis, we then statistically tested for a main effect of state (threat-of-shock or safety) at both the encoding and retrieval stages, using a within-subjects factorial analysis of variance (ANOVA) of the extracted beta coefficients.” – Main Manuscript, page 11

“To address our secondary neural hypothesis that hippocampus activation is increased while retrieving faces under the same state (threat-of-shock or safety) as they were encoded in, we used the resampled bilateral hippocampus mask (using AFNI’s 3dmaskave) previously extracted by Garibbo et al., (2019) from the Wake Forest University PickAtlas toolbox (Maldjian et al., 2003) as an ROI for the following contrast of interest: faces during state congruent retrieval blocks (threat-of-shock/safety at encoding followed by threat-of shock/safety at retrieval) > incongruent retrieval blocks (threat-of-

shock/safety at encoding followed by safety/threat-of shock at retrieval). To ensure consistency with the behavioural analysis, we then statistically tested for a main effect of state congruency using a within-subjects factorial analysis of variance (ANOVA) of the extracted beta coefficients.” – Main Manuscript, page 11

“As for the behavioural data, we followed up on the above primary and secondary neural analyses with a ROI mega-analysis. We did this by combining the beta weight coefficients from the within-subject modelling of our current sample with those available from Garibbo et al. (2019) for a group-level analysis (total n=124) for the contrasts of interest and ROIs specified in the aforementioned neural hypotheses. We then statistically tested for the same main effects using ANOVA models, as in our behavioural mega-analysis, by accounting for study as a between-subjects factor.” – Main Manuscript, page 11

“We followed up on the encoding effect with a whole-brain mega-analysis by combining the beta weights from the within-subject modelling of our sample with those available from Garibbo et al. (2019) for a group-level analysis (total n=124) of the same contrast of interest: faces during threat-of-shock blocks > safety blocks, for encoding. This was done to determine if both the anterior cingulate (ACC) cluster identified by Garibbo et al. (2019) and the posterior cingulate cortex (PCC) cluster identified in our primary whole-brain analysis would emerge in the combined sample (voxel-wise threshold = $p < 0.001$, cluster-level significance threshold of $p < 0.05$). Note that to avoid double dipping we did not use the mega-analysis for the identification and discussion of new clusters. We specified the group-level ANOVA model (using AFNI’s 3dMVM) in the same manner as in our behavioural mega-analysis, by accounting for study as a between-subjects factor.” – Main Manuscript, page 12

Results:

In the anterior cingulate cortex (ACC) ROI in our sample we found no significant difference in neural activation when comparing threat-of-shock to safety during encoding ($F=1.208$, $df=91$, $p=0.275$, $\eta_p^2=0.013$) or retrieval ($F=1.416$, $df=91$, $p=0.237$, $\eta_p^2=0.015$). However, the whole brain analysis of our sample revealed significant activation when comparing threat-of-shock to safety during encoding, but not retrieval, in a posterior cingulate cortex (PCC) cluster (size: 43 voxels, peak:

$x=+0.5$, $y=+38.5$, $z=+49.5$, centre of mass: $x=+0.4$, $y=+32.7$, $z=+44.3$, see figure 2a for group-level cluster and supplementary figure 3a for visualisation of coefficients).

We then followed up on this with an individual participant data (IPD) exploratory mega-analysis of the ACC ROI and whole-brain to determine what neural activation emerges in a more powered analysis across samples. For this we combined the beta weights from the within-subject models for the same contrast of interest from Garibbo et al. (2019) with our sample (total $N=123$), accounting for study as a between-subjects factor. In the ACC ROI, there was a significant effect of encoding state, with increased activation when faces were encoded under threat-of-shock compared to safety ($F=55.921$, $df=121$, $p < 0.001$, $\eta_p^2=0.316$) but not retrieval state ($F=0.371$, $df=121$, $p = 0.544$, $\eta_p^2=0.003$) in the combined sample. The whole-brain analysis further revealed that both the PCC cluster (size: 274 voxels, peak: $x=-1.5$, $y=+28.5$, $z=+43.5$, centre of mass: $x=+3.7$, $y=+27.1$, $z=+43.8$) and ACC cluster (size: 1009 voxels, peak: $x=+0.5$, $y=-45.5$, $z=+21.5$, centre of mass: $x=+1.7$, $y=-45$, $z=+17.8$) emerged when contrasting threat-of-shock with safe face encoding, but not retrieval, in the combined sample at a voxel-wise threshold of $p<0.001$ and cluster-level significance threshold of $p<0.05$ (see figure 2b for group-level cluster and supplementary figure 3b,c for visualisation of coefficients).

Neural Effects of State Congruency at Retrieval in Hippocampus

Using an ROI analysis to address our secondary neural hypothesis, we found evidence for a significant increase in average neural activation in the bilateral hippocampus when the state (threat-of-shock or safety) during retrieval was congruent with encoding (see figure 2c for group-level ROI cluster and supplementary figure 4b,c for visualisation of coefficients). This was reported in the original study (see Garibbo et al., 2019) and replicated in our sample ($F=7.144$, $df=91$, $p=0.009$, $\eta_p^2=0.073$), as well as in a combined ROI mega-analysis ($F=13.534$, $df=121$, $p<0.001$, $\eta_p^2=0.101$).

– Main Manuscript, page 15-16

MINOR / SPECIFIC

50. the authors could consider tightening up Figure 2 my reducing white space

In line with this and other reviewer comments we have now changed figure 2 to improve the visualisation of our main fMRI results and reduce the white space. See below:

Figure 2. Neural Activation during the threat-of-shock face recognition task showing that A) a posterior cingulate cortex cluster (PCC; see thresholded group-level cluster with t-statistic bar) was significantly more active when encoding faces under threat-of-shock compared to safety in a whole-brain analysis of the current study. Further, when combining the current study and within-subject results from Garibbo et al. (2019) in a whole brain mega-analysis, we found that B) significantly enhanced neural activation while encoding faces under threat-of-shock compared to safety was evident in both the anterior cingulate cortex (ACC) and posterior cingulate cortex (PCC). In addition, congruency in the state (threat-of-shock or safety) between the encoding and retrieval phase was associated with C) increased activation in a bilateral hippocampus ROI when the state during retrieval was congruent compared to incongruent with encoding in the current study (Buehler et al. 2023: n=92) and a mega-analysis combining the current study with Garibbo et al. (2019). See supplementary figure 3a,b,c for visualisation of the model coefficients.

– Main Manuscript, page 16

The additional plots showing the significant model coefficients can now be found in the supplementary materials. See below:

Neural Activation underlying Face Encoding under Threat-of-Shock

Supplementary Figure 3. Beta coefficients extracted for the significant model contrasts from the whole-brain analysis of neural activation showing that in the current study A) there was a significant effect in the posterior cingulate cortex cluster (PCC) encoding of faces under threat-of-shock compared to safety. Further, when combining the current study and within-subject results from Garibbo et al. (2019) in a whole brain mega-analysis, there was a significant effect while encoding faces under threat-of-shock compared to safety in both D) the anterior cingulate cortex (ACC) and E) posterior cingulate cortex (PCC). Note the beta coefficients in plot B), D) and E) were extracted from the significant whole-brain group-level contrasts shown in the main results figure 2 and plotted here for visualisation purposes only.

Neural Effects of State Congruency at Retrieval

Supplementary figure 4. Beta coefficients for the state-congruency contrast extracted from the significant A) bilateral hippocampus ROI and showing increased activation in the hippocampus when the state during retrieval is congruent compared to incongruent with encoding in A) the current study (Buehler et al. 2023: n=92) and B) a mega-analysis combining the current study with Garibbo et al. (2019).

– Supplementary Material, page 3

51. “To determine the spatial specificity of this effect, we used the ACC activation from Garibbo et. al as a region of interest (ROI) alongside a whole-brain analysis, since cingulate regions may form part of a wider neural network involved in anxiety processing (Chavanne & Robinson, 2021).”

As written, there is no logical connection between using the ACC ROI from the previous paper and determining spatial specificity. What do you mean?

We agree and have therefore removed claims of spatial specificity. This section of the introduction now reads as follows:

“We therefore used the ACC activation from Garibbo et. al as a region of interest (ROI) alongside a whole-brain analysis” – Main Manuscript, page 4

52. “Before data analysis, the first four volumes were discarded to allow the magnetic field to stabilize.”

-this sentence should be moved to earlier in the paragraph

Done!

53. “The first 4 dummy volumes were discarded to allow the magnetic field to stabilize.”

-this sentence is redundant with the later sentence “Before data analysis, the first four volumes were discarded to allow the magnetic field to stabilize.” One of these can be dropped.

Thank you for noticing, we dropped the latter.

54. “To make inferences about the neural activation underlying the threat-of-shock potentiated face recognition task...”

-is “potentiated” the most appropriate term here? “Attenuated” may be more appropriate based on directionality of the effect

We agree with the reviewer that potentiated may not be the most appropriate term, but we also want to avoid using attenuated in the task description to avoid implying that threat-of-shock generally has an attenuating effect on face recognition in this task. To keep it neutral we now removed potentiated and instead refer to the task simply as “threat-of-shock face recognition task”.

55. The authors may want to consider the differences between incidental vs. integral emotion. For instance, imagine that you are encoding faces while being threatened with shock, as in the present work. How might this differ from more task-relevant anxiety, such as when the faces being encoded are the threat?

For further details: Shackman, A. J., Sarinopoulos, I., Maxwell, J. S., Pizzagalli, D. A., Lavric, A., & Davidson, R. J. (2006). Anxiety selectively disrupts visuospatial working memory. *Emotion*, 6(1), 40.

We thank the reviewer for this comment, which is an important distinction that we fully agree with, and have thus integrated this consideration into our discussion as follows:

“In line with other work on dual process competition it is specifically in the case of task-irrelevant anxiety, which we were interested in eliciting by our threat-of-shock paradigm, that arousal signals

compete with cognitive processes, so this may be reversed when the source of threat is task-relevant (Lago et al., 2022; Robinson et al., 2013; Shackman et al., 2006). “– Main Manuscript, page 17

56. Consider including basic demographic data for participants, both within and across samples (mean age? Percent female?)

We have now added this to the participants subsection of the methods for all samples:

“We collected data for a total sample of n=98 participants, who completed the threat-of-shock face recognition task during a single fMRI scanning session between December 2017 and May 2022. This sample forms part of a larger pharmacological intervention study, but here we utilize only the data from asymptomatic individuals at baseline. Of those 98 individuals, five did not complete the task of interest so the final sample size for this study was n=93 participants (66 females, 27 males, mean age=24.22). – Main Manuscript, page 5

For additional meta-analyses, we also collated data from two studies. These included n=86 participants (50 females, 36 males, mean age=24.7) who had completed the threat-of-shock face recognition task during a behavioural testing session (see Bolton & Robinson, 2017) and n=32 participants (18 females, 14 males, mean age=27.03) who had completed it during a fMRI scanning session (see Garibbo et al., 2019).” – Main Manuscript, page 5

57. as I commented on with respect to the introduction, what type of biomarker this work would contribute to is unclear in this context. Additionally, if the biomarker identifies face recognition deficits on the basis of brain activation in a threat-of-shock task, it is clear that there is any interest among clinicians or scientists in developing such a biomarker. It also merits comments that it is not guaranteed that a biomarker developed in the present sample would have any relevance to patients with anxiety disorders.

Thank you for this feedback. Please see our changes to remove these implications for biomarker development in response to previous reviewer comments.

58. ”For the within-subject modeling we constructed general linear models (GLMs). With the regressors of interest we accounted for both preceding and subsequent state, threat-of-shock (TH) or safety (SF), in our contrasts of encoding and retrieval by including the face stimuli onset times for all combinations of state at encoding (encoding face onsets: THTH, THSF, SFTH, SFSF) and retrieval (retrieval face onsets: THTH, THSF, SFTH, SFSF). Then we also included the contrasts for TH encoding > SF encoding and TH retrieval > SF retrieval. These were convolved with the hemodynamic response function (approximated by a gamma function) using ‘3dDeconvolve’ in AFNI. As nuisance regressors in all within-subject GLMs we included movement-correction parameters

(pitch, roll, yaw, z, y, z and derivatives of each motion type) as well as a parameter controlling for the time of shock delivery and presentation time of the safe/threat warnings.”

Did you model it as a block or as a series of events (e.g., one stick per face, but it’s basically equivalent to a block since the events are so close together)? It is not entirely clear as written.

We thank the reviewer for this question and apologize if this was unclear. We confirm that face stimuli onset times were modelled as events in the first-level within-subject models, but since the events are so close together our main analyses and inferences at the group-group level used block contrasts.

To make this clear we have now rewritten the within subject modelling subsection of the methods as follows:

“For the within-subject modelling we constructed general linear models (GLMs) for each participant. With the regressors of interest we modelled the face onset times and duration (0.5 seconds) as events, with the fixation cross during the inter-stimulus intervals as well as start and end of task treated as an implicit baseline. This amounted to 8 regressors accounting for the face stimuli onsets during all state combinations, including 4 encoding phase regressors with 2 for stimuli during threat-of-shock encoding (i.e., once followed by threat-of-shock and once by safety at retrieval) and 2 for safety encoding (i.e., once followed by threat-of-shock and once by safety at retrieval), as well as 4 retrieval phase regressors for the same combinations of state but with stimuli onsets during retrieval. These were convolved with the hemodynamic response function (approximated by a gamma function) using ‘3dDeconvolve’ in AFNI. As nuisance regressors in all within-subject GLMs we further included movement-correction parameters (pitch, roll, yaw, z, y, z and derivatives of each motion type) as well as a parameter controlling for the time of shock delivery, presentation time of the safe/threat-of-shock warnings and response slide. To further control for motion artifacts, we censored volumes with framewise displacement exceeding 1.3 mm and excluded individuals with more than 20% of volumes requiring censoring (none in this sample).” – Main Manuscript, page 10

We have also adjusted the relevant group-level modelling subsections of the methods in the following manner to clarify that we used blocks as our contrasts of interest:

“To address our primary neural hypothesis that threat-of-shock at encoding but not retrieval is associated with increased ACC activation, we used the resampled group-level dorsal ACC activation map (using AFNI’s 3dmaskave) from Garibbo et al., (2019) as an ROI for the following contrast of interest: threat-of-shock blocks > safety blocks, for encoding and retrieval separately.” – Main Manuscript, page 11

59. the authors should consider controlling for study/pipeline since they seem to be confounded across samples

We thank the reviewer for this suggestion and now control for study in all our analyses of the combined samples (across studies). This was already controlled for in our behavioural mega-analysis and we have now included the same control for study as a between-subjects factor in our fMRI whole-brain mega-analysis to make this consistent. We note that the inference has remained the same, but we have updated the methods and results section accordingly:

Methods Section:

“Exploratory Mega-Analyses of Behaviour across studies

We also performed a mega-analysis, considered the ‘gold-standard’ of meta-analytic approaches and also referred to as individual-participant data (IPD) meta-analysis (Eisenhauer, 2021; Tierney et al., 2015). For this we combined the raw behavioural data from all participants (n=210), across the current study (referred to as Buehler et al. 2023: n=92) and the two previous studies utilizing the same threat-of-shock potentiated face recognition task (Bolton & Robinson, 2017: n=86; Garibbo et al., 2019: n=32). Based on the aforementioned behavioural hypotheses, this was done to assess the robustness of a) the effect of state (threat-of-shock or safety) distinctly at encoding and retrieval, as well as b) the effect of state congruency at retrieval on face recognition accuracy across all three studies. We therefore used the same ANOVA models as in the primary and secondary analyses but accounting for study as a between-subjects factor. Of interest were the main effects of encoding state and retrieval state on recognition accuracy.” – Main Manuscript, page 9

Exploratory Mega-Analyses of ROIs across studies

As for the behavioural data, we followed up on the above primary and secondary neural analyses with a ROI mega-analysis. We did this by combining the beta weight coefficients from the within-subject modelling of our current sample with those available from Garibbo et al. (2019) for a group-level analysis (total n=124) for the contrasts of interest and ROIs specified in the aforementioned neural hypotheses. We then statistically tested for the same main effects using ANOVA models, as in our behavioural mega-analysis, by accounting for study as a between-subjects factor. – Main Manuscript, page 11

Exploratory Mega-Analyses of whole-brain Activation across studies

We followed up on the encoding effect with a whole-brain mega-analysis by combining the beta weights from the within-subject modelling of our sample with those available from Garibbo et al. (2019) for a group-level analysis (total n=124) of the same contrast of interest: faces during threat-of-shock blocks > safety blocks, for encoding. This was done to determine if both the anterior

cingulate (ACC) cluster identified by Garibbo et al. (2019) and the posterior cingulate cortex (PCC) cluster identified in our primary whole-brain analysis would emerge in the combined sample (voxel-wise threshold = $p < 0.001$, cluster-level significance threshold of $p < 0.05$). Note that to avoid double dipping we did not use the mega-analysis for the identification and discussion of new clusters. We specified the group-level ANOVA model (using AFNI's 3dMVM) in the same manner as in our behavioural mega-analysis, by accounting for study as a between-subjects factor. – Main Manuscript, page 12

Results Section:

“We then ran an individual participant data (IPD) mega-analysis across the behavioural data from the two previous studies (Bolton & Robinson, 2017: $n=86$; Garibbo et al., 2019: $n=32$) and current study (referred to as Buehler et al. 2023: $n=92$), to determine what effects are robust in a combined sample ($n=211$). This revealed a significant main effect of state on face recognition for encoding ($F=16.777$, $df=207$, $p < 0.001$, $\eta_p^2=0.075$) but not retrieval ($F=3.060$, $df=207$, $p=0.082$, $\eta_p^2=0.015$), while accounting for study as a between-subjects factor.” – Main Manuscript, page 13

“Neural Activation underlying Face Encoding under Threat-of-Shock

In the anterior cingulate cortex (ACC) ROI in our sample there was no significant difference in neural activation when comparing threat-of-shock to safety during encoding ($F=1.208$, $df=91$, $p=0.275$, $\eta_p^2=0.013$) or retrieval ($F=1.416$, $df=91$, $p=0.237$, $\eta_p^2=0.015$). However, the whole brain analysis of our sample revealed significant activation when comparing threat-of-shock to safety during encoding, but not retrieval, in a posterior cingulate cortex (PCC) cluster (size: 43 voxels, peak: $x=+0.5$, $y=+38.5$, $z=+49.5$, centre of mass: $x=+0.4$, $y=+32.7$, $z=+44.3$, see figure 2a for group-level cluster and supplementary figure 3a for visualisation of coefficients).

We then followed up on this with an exploratory mega-analysis of the ACC ROI and whole-brain to determine what neural activation emerges in a more powered analysis across samples. For this we combined the beta weights from the within-subject models for the same contrast of interest from Garibbo et al. (2019) with our sample (total $N=123$), accounting for study as a between-subjects factor. In the ACC ROI, there was a significant effect of encoding state, with increased activation when faces were encoded under threat-of-shock compared to safety ($F=55.921$, $df=121$, $p < 0.001$, $\eta_p^2=0.316$) but not retrieval state ($F=0.371$, $df=121$, $p = 0.544$, $\eta_p^2=0.003$) in the combined sample. The whole-brain analysis further revealed that both the PCC cluster (size: 274 voxels, peak: $x=-1.5$, $y=+28.5$, $z=+43.5$, centre of mass: $x=+3.7$, $y=+27.1$, $z=+43.8$) and ACC cluster (size: 1009 voxels,

peak: $x=+0.5$, $y=-45.5$, $z=+21.5$, centre of mass: $x=+1.7$, $y=-45$, $z=+17.8$) emerged when contrasting threat-of-shock with safe face encoding, but not retrieval, in the combined sample at a voxel-wise threshold of $p<0.001$ and cluster-level significance threshold of $p<0.05$ (see figure 2b for group-level cluster and supplementary figure 3b,c for visualisation of coefficients).

Neural Effects of State Congruency at Retrieval in Hippocampus

Using an ROI analysis to address our secondary neural hypothesis, we found evidence for a significant increase in average neural activation in the bilateral hippocampus when the state (threat-of-shock or safety) during retrieval was congruent with encoding (see figure 2c for group-level ROI cluster and supplementary figure 4b,c for visualisation of coefficients). This was reported in the original study (see Garibbo et al., 2019) and replicated in our sample ($F=7.144$, $df=91$, $p=0.009$, $\eta_p^2=0.073$), as well as in a combined ROI mega-analysis ($F=13.534$, $df=121$, $p<0.001$, $\eta_p^2=0.101$).”

2nd Jul 24

Dear Ms Buehler,

Your manuscript titled "Investigating the replicability of neural mechanisms underlying anxiety-attenuated face encoding" has now been seen by our reviewers, whose comments appear below. In light of their advice I am delighted to say that we are happy, in principle, to publish a suitably revised version in *Communications Psychology* under the open access CC BY license (Creative Commons Attribution v4.0 International License).

We therefore invite you to revise your paper one last time to address the remaining concerns of our reviewers and a list of editorial requests. At the same time we ask that you edit your manuscript to comply with our format requirements and to maximise the accessibility and therefore the impact of your work.

EDITORIAL REQUESTS:

In response to our previous editorial requests, you toned down claims of specificity (as there is no positive evidence for a null effect at retrieval and an absence of a significant interaction in some contrasts). What is missing at this stage is a more comprehensive report of interaction contrasts, including in the fMRI analyses. It is necessary to report these regardless of the appropriate absence of claims of specificity (our guidelines require comprehensive statistics reporting, including non-significant findings). I have detailed some instances in the attached editorial checklist. Moreover, please note that the code sharing for fMRI analyses as currently described in the Code Availability Statement is not compliant with our guidelines. I included more information in the Editorial Request Table.

SUBMISSION INFORMATION:

OPEN ACCESS:

Communications Psychology is a fully open access journal. Articles are made freely accessible on publication under a CC BY license (Creative Commons Attribution 4.0 International License). This license allows maximum dissemination and re-use of open access materials and is preferred by many research funding bodies.

For further information about article processing charges, open access funding, and advice and support from Nature Research, please visit <https://www.nature.com/commspsychol/article-processing-charges>

At acceptance, you will be provided with instructions for completing this CC BY license on behalf of all authors. This grants us the necessary permissions to publish your paper. Additionally, you will be asked to declare that all required third party permissions have been obtained, and to provide billing information in order to pay the article-processing charge (APC).

* DATA AVAILABILITY:

[link redacted]

Best regards,

Marike, on behalf of

Xiaoqing Hu

Marike Schiffer, PhD

Chief Editor

Communications Psychology

Xiaoqing Hu, PhD

Editorial Board Member
Communications Psychology
orcid.org/0000-0001-8112-9700

REVIEWERS' COMMENTS:

Reviewer #1 (Remarks to the Author):

The authors have addressed my comments adequately. I have no further comments.

Reviewer #4 (Remarks to the Author):

#####

Investigating the replicability of neural mechanisms underlying anxiety-attenuated encoding of emotional faces

Buehler...& Robinson

#####

The present study leverages a relatively large sample to replicate and extend prior work on the impacts of threat-of-shock-induced anxiety on the encoding and recognition stage of emotional face processing. The authors conclude that anxiety impairs face recognition, possibly due to attentional interference during encoding, and that the cingulate cortex is critical to this process.

Major

The authors have done a thorough and thoughtful job of revising the manuscripts to address all previous major concerns. Nice work!

Minor

To our knowledge, only two previous studies investigated behaviourally (Bolton & Robinson, 2017; Garibbo et al., 2019) as well as using functional magnetic resonance imaging (fMRI) (Garibbo et al., 2019) how a within-subjects threat-of-shock anxiety induction distinctly affects the encoding and retrieval stage of emotional face processing.

-This sentence is a bit hard to follow as currently worded - you might consider rewording for clarity. As an option, "To our knowledge, only two previous studies employed behavioral (Bolton & Robinson, 2017; Garibbo et al., 2019) and functional magnetic resonance imaging (fMRI) (Garibbo et al., 2019) to investigate how a within-subjects threat-of-shock anxiety induction distinctly affects the encoding and retrieval stage of emotional face processing."